# STEERING LARGE LANGUAGE MODELS BETWEEN CODE EXECUTION AND TEXTUAL REASONING

**Yongchao Chen**
MIT / Harvard
yongchaochen@fas.harvard.edu

**Harsh Jhamtani**
Microsoft
hjhamtani@microsoft.com

**Srinagesh Sharma**
Microsoft
srsharm@microsoft.com

**Chuchu Fan**
MIT
chuchu@mit.edu

**Chi Wang**
Google DeepMind
chi@chiwang.cc

## ABSTRACT

While a lot of recent research focuses on enhancing the textual reasoning capabilities of Large Language Models (LLMs) by optimizing the multi-agent framework or reasoning chains, several benchmark tasks can be solved with 100% success through direct coding, which is more scalable and avoids the computational overhead associated with textual iterating and searching. Textual reasoning has inherent limitations in solving tasks with challenges in math, logics, optimization, and searching, which is unlikely to be solved by simply scaling up the model and data size. The recently released OpenAI GPT Code Interpreter and multi-agent frameworks such as AutoGen have demonstrated remarkable proficiency of integrating code generation and execution to solve complex tasks using LLMs. However, based on our experiments on 7 existing popular methods for steering code/text generation in both single- and multi-turn settings with 14 tasks and 6 types of LLMs (including the new O1-preview), currently there is no optimal method to correctly steer LLMs to write code when needed. We discover some interesting patterns on when models use code vs. textual reasoning with the evolution to task complexity and model sizes, which even result in an astonishingly inverse scaling behavior. We also discover that results from LLM written code are not always better than using textual reasoning, even if the task could be solved through code. To mitigate the above issues, we propose three methods to better steer LLM code/text generation and achieve a notable improvement. The costs of token lengths and runtime are thoroughly discussed for all the methods. We believe the problem of steering LLM code/text generation is critical for future research and has much space for further improvement. Project Page, Datasets, and Codes are available at https://yongchao98.github.io/CodeSteer/.

## 1 INTRODUCTION

The rapid progress of LLMs has inspired a great quantity of research in building general language-guided agents that can solve various tasks automatically (Wu et al., 2023; Li et al., 2023a; Yao et al., 2024; Besta et al., 2024). While the capabilities of these LLM-based agents have been largely enhanced by tuning the agent frameworks (Momennejad et al., 2024; Wang et al., 2024a), reasoning chains (Li et al., 2023a; Yao et al., 2024; Besta et al., 2024), visual and spatial abilities (Zhai et al., 2024; Yuan et al., 2024), and input prompts (Suzgun & Kalai, 2024; Chen et al., 2024b; Cheng et al., 2023), the best LLMs still make mistakes on simple tasks (Zhou et al., 2024), such as the recently popular topics of '9.11' and '9.9' numerical comparison and 'r' letter count in 'strawberry', as shown in Fig 1. However, after reviewing all the tested tasks from previous papers, we detect that nearly half of the tasks can be completely solved by coding, such as Blocksworld (Valmeekam et al., 2024), Game 24 (Zhou et al., 2023a), and Logical Deduction (Suzgun et al., 2022).

Text is suitable for semantic analysis and commonsense reasoning, but is not the best format for precise computation and planning, symbolic manipulation, and algorithmic processing and opti-

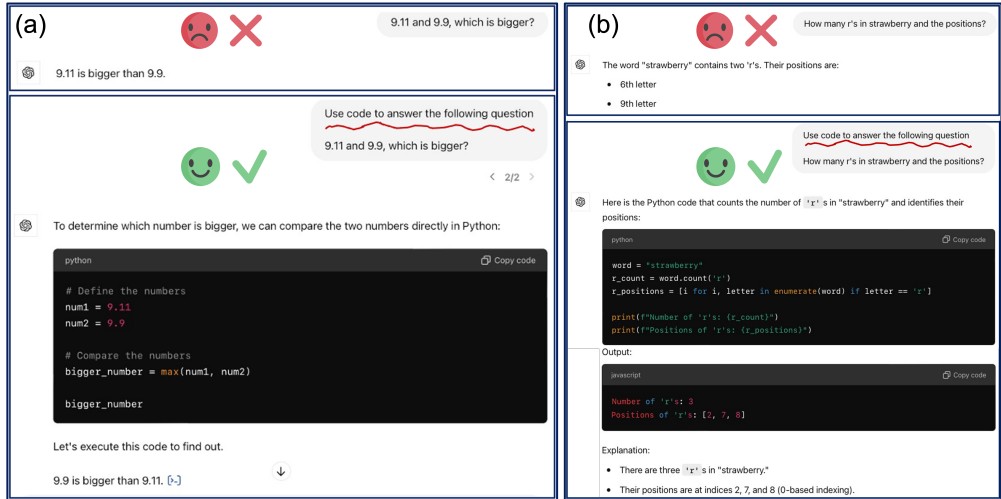

Figure 1: The cases that GPT-4o makes simple mistakes by direct textual reasoning but can reliably solve the problem with prompted to use code.

mization (Kambhampati et al., 2024b; Chen et al., 2024a). Conversely, programs excel in rigorous operations, and can outsource intricate calculations to specialized tools like equation solvers. Since recent LLMs are well trained at code generation (Bairi et al., 2024), one question that comes up is whether querying LLMs to generate code can be more effective than textual reasoning.

In this study, we emphasize that textual reasoning has inherent limitations in solving tasks that involve math, logic, and optimization, where coding can often provide a better solution. For example, Fig 1 presents two typical examples that the ChatGPT of GPT-4o makes mistakes by direct textual reasoning but easily solves the problem after prompted to use code. Recent studies also show that using a code-based framework enhances LLMs' logical reasoning performance, even in commonsense reasoning tasks (Madaan et al., 2022; Liang et al., 2022; Chen et al., 2022). Targeting text as the only output modality has limitations in the scalability of task complexity. As shown in Appendix Fig 9, even if the model of OpenAI series scales from GPT-3.5 to O1, LLM still easily fails once the task complexity grows. In these tasks, coding is a scalable and complete solution.

Guiding LLMs to choose between code generation/execution and textual reasoning remains a challenging problem, as common questions lack prior cues for either approach. Recent OpenAI GPT models address this issue by augmenting the Code Interpreter (CI) function, where models are trained to use an integrated coding platform as part of their reasoning (Achiam et al., 2023; Dubey et al., 2024). Once the model generates code, the platform executes it and returns the results for further processing. This iterative process of generating code and text continues until the final answer is reached. In addition to GPT CI, multi-agent frameworks like AutoGen (Wu et al., 2023) query the LLM itself, using a specific system prompt to decide when to generate code, which is then executed based on predefined rules. More related work are specifically discussed in Appendix Section A.

In this paper, we perform an in-depth investigation into the effectiveness of LLMs in steering between use of textual reasoning and code generation/execution across 14 diverse tasks requiring mathematical, verbal, and planning capabilities, using 6 types of LLMs (O1-preview, GPT-4o (Achiam et al., 2023), GPT-4o-mini, GPT-3.5 (Brown, 2020), Claude-sonnet (Anthropic, 2024), Mixtral-8x7b (Jiang et al., 2024)). The key contributions and findings of our work are:

1. **Existing methods struggle to optimally decide when to write code or use textual reasoning**: We evaluated 10 different methods to steer LLMs to use code when required by conducting experiments across 14 datasets/tasks and 6 LLMs (with and without built-in CI, prompt modifications to favor code over text, multi-turn code refinement, etc.). Our experiments reveal that there is no single best method across the board. Additionally, we also analyze the trade-offs in token length and runtime against accuracy for each method, which can serve as a guidance to pick the method of choice based on budget and performance expectations.

2. **Forcing LLMs to write code is not guaranteed to give more accurate results compared to textual reasoning:** Our experiments suggest that prompting LLM to answer directly with code can sometimes lead to worse overall accuracy. We discuss several contributing reasons for such behavior. Firstly, writing correct code is tough in certain tasks, such as robot task planning. Secondly, code format can limit the space of output tokens, potentially hindering the reasoning ability of LLMs (Tam et al., 2024). Moreover, we observe that LLMs sometimes generate code that resembles more of textual reasoning rather than containing any functional implementations.

3. **Our experiments reveal the patterns on when LLMs use code vs. textual reasoning as a function of factors such as task complexity and model sizes:** Surprisingly, when augmented with CI, smaller models like GPT-3.5 sometimes outperform larger ones like GPT-4o, illustrating an inverse scaling behavior, contrary to previous studies (Kaplan et al., 2020). This phenomenon appears to be linked to the varied LLM's confidence in its textual reasoning ability. As a result, GPT-3.5 outperforms GPT-4o in Game 24 and Number Multiplying tasks (inverse scaling behavior).

4. **Mixing code and textual reasoning, and multi-turn refinement:** Inspired by previous work that utilize multi-agent framework to refine answers (Chen et al., 2023; Wang et al., 2024a), we propose optimized methods like assembling coding and textual reasoning together resulting into improvements across 6 models. We also show that multi-turn execution/refinement (Chen et al., 2024a; Kambhampati et al., 2024a) improves the performance. However, we also find that these methods are limited by the LLM's inherent capabilities and the frequency of code usage, which in turn increases runtime costs and token lengths.

## 2 OPENAI GPT CODE INTERPRETER STRUGGLES IN CODE/TEXT CHOICES

Are current methods capable of effectively switching between code execution and textual reasoning? Since OpenAI GPT CI is the current most popular and effective method for steering code/text generation (Wang et al., 2023a; Zhou et al., 2023b), we carry out a detailed exploration on its characteristics and limitations. In this section, we use Number Multiplying (calculating number multiplication) and Game 24 (outputting an equation that evaluates to 24 with the given set of integers) as representative tasks because they are simple to describe to LLMs without being affected by different prompt types, and their complexity can be easily adjusted. The question prompt is the same as the original dataset (Zhou et al., 2023a; Yao et al., 2024) without the extra hints for code/text steering.

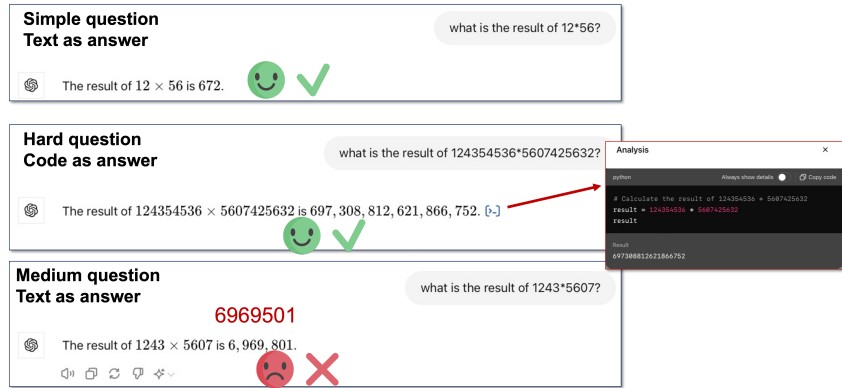

Figure 2: GPT-4o CI tends to handle simple Number Multiplying tasks with text and complex tasks with code. However, it often fails with medium-difficulty questions, where it is overconfident and chooses not to use code when needed.

### 2.1 EVOLUTION WITH TASK COMPLEXITY

We observe an intriguing property of GPT CI: its decision to use code depends on the complexity of the task, as shown in Fig 2. GPT-4o CI chooses to handle simple Number Multiplying questions with text and complex questions with code, resulting in correct answers. However, it fails in medium-difficulty questions since it tends to be overconfident and chooses to answer the question via textual

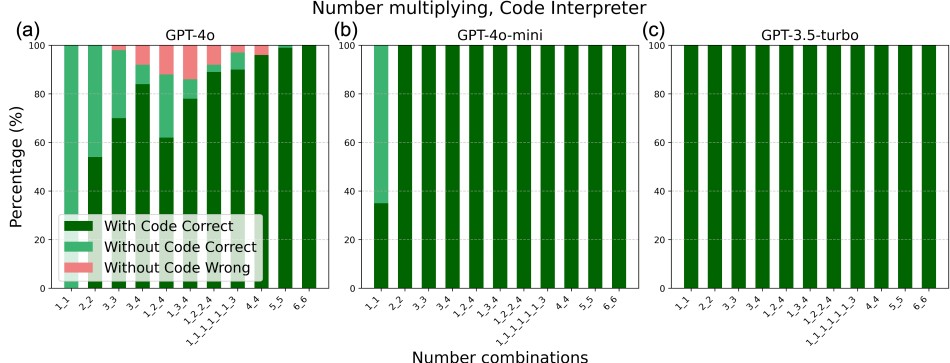

Figure 3: Success rates and code usage rates of GPT CI in Number Multiplying task across varied task complexity. The labels on the x-axis represent the number of digits in the numbers being multiplied. For example, '3_4' means a three-digit number multiplied by a four-digit number. From the left to the right of x-axis, the digit numbers of multiplied values increase, representing increasing task complexity. The success and failure cases are visualized with green and red colors, respectively.

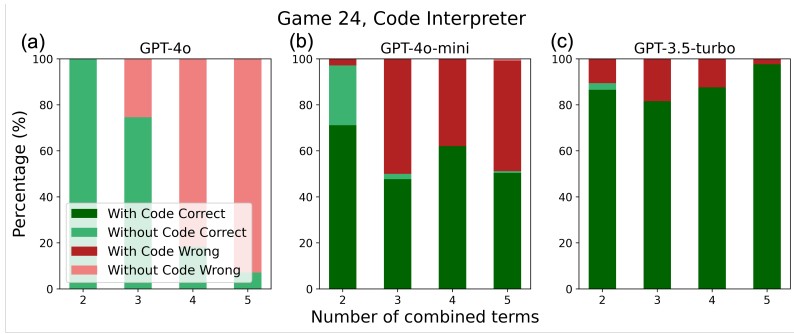

Figure 4: Success rates and code usage rates of OpenAI CI in Game 24 task across varied task complexity. The labels on the x-axis represent the number of terms/values to be combined to form value 24, presenting an increasing task complexity from the left to the right side. The success and failure cases are visualized with green and red colors, respectively.

reasoning, which sometimes is wrong. Fig 3a visualizes this phenomenon quantitatively. We adjust the task complexity via changing the number of digits in the numbers being multiplied. GPT-4o answers all multiplication questions correctly using text for small numbers (left side) and generates code for very large numbers (right side). Errors occur when the numbers are neither too large nor too small, causing GPT-4o to struggle with deciding whether to use code. In the Number Multiplying task, generating code is straightforward, leading to correct answers when code is used. However, GPT-4o's tendency to be overconfident and avoid using code in medium-difficulty questions results in occasional failures.

## 2.2 INVERSE SCALING BEHAVIOR OF MODEL SIZES

Based on the above property, we also see that in some tasks smaller models outperform larger models when all augmented with CI, which is inverse to the well-known scaling law in LLMs (Kaplan et al., 2020). In Fig 3, both GPT-4o-mini and GPT-3.5 achieve 100% success rates across all the task complexity. Compared to GPT-4o, GPT-4o-mini and GPT-3.5 are more conservative so that they generate code all the time when encountering slightly complex questions. This phenomenon also appears in Game 24 task (Fig 4) where GPT-4o generates text in all the questions while GPT-4o-mini and GPT-3.5 generate code in most cases. In Game 24, coding is also a more reliable method than textual reasoning. Hence, GPT-4o-mini and GPT-3.5 notably outperform GPT-4o, especially when task complexity is high. In the Game 24 task, both GPT-4o-mini and GPT-3.5 tend to generate code more frequently as task complexity increases, further supporting the conclusion in Section 2.1.

We hypothesize that GPT-4o will increasingly use code as task complexity rises. However, our additional tests (not included in the paper) indicate that GPT-4o only begins using code to solve Game 24 problems when the complexity of the tasks becomes exceedingly high.

Furthermore, in Fig 4, we observe that GPT-3.5 outperforms GPT-4o-mini, even though both models primarily generate code in most test cases. To reveal the underlying mechanism, Fig 5 shows a typical correct code response from GPT-3.5 compared to an incorrect one from GPT-4o-mini. Although the LLM CI truly uses code generation/execution, in some cases, the code serves merely as a form of textual reasoning, calculating and verifying values without efficiently aiding in finding the correct answer, unlike the correct answer version. The reason GPT-4o-mini CI generates text-like code answers more frequently than GPT-3.5 CI may stem from its greater confidence in its textual reasoning abilities. This overconfidence leads to an inverse scaling effect, despite both LLMs employing code.

Based on the discussion, we conclude that larger models in the GPT CI series tend to rely more on their textual reasoning abilities, often becoming overconfident and opting not to use code, even when coding would be more effective. When they do use code, it often serves as a supplement to textual reasoning rather than the primary problem-solving tool. This results in the phenomenon that smaller GPT CI models perform better than larger ones in some tasks, i.e., the inverse scaling behavior.

```
Correct code answer by GPT-3.5
Code Interpreter

assistant > code_interpreter

from itertools import permutations

def find_equation(numbers):
    for perm in permutations(numbers):
        ops = ['+', '-', '*', '/']
        for op1 in ops:
            for op2 in ops:
                for op3 in ops:
                    eq = f"(({perm[0]} {op1} {perm[1]})
{op2} {perm[2]}) {op3} {perm[3]}"
                    try:
                        if eval(eq) == 24:
                            return eq
                    except ZeroDivisionError:
                        pass
    return "No valid equation found"

numbers = [1, 4, 6, 12]
equation = find_equation(numbers)
equation
assistant > The equation that satisfies the conditions
and evaluates to 24 is: ((1 * 6) - 4) * 12.
```

```
Wrong code answer by GPT-4o-mini Code Interpreter

assistant > code_interpreter

# Let's calculate the expression for the input numbers 1, 4, 6, and 12 to obtain 24.
# Given numbers
numbers = [1, 4, 6, 12]

# Trying to find a valid expression that equals 24
# One possible expression is: (6 * 4) + (12 - 1) = 24
expression = (6 * 4) + (12 - 1)
expression
assistant > It seems that my initial attempt did not yield the correct result. Let's try to
find a valid expression that equals 24 using the numbers 1, 4, 6, and 12.

After some consideration, one valid expression is:
\[
(12 - 6) \times 4 + 1 = 24
\]

Let's verify this calculation.
# Verifying the expression (12 - 6) * 4 + 1
result = (12 - 6) * 4 + 1
result
assistant > The expression I provided also did not yield 24. Let's try a different approach
to find a valid expression using the numbers 1, 4, 6, and 12 to achieve the target of 24.

One possible expression is:
\[
(6 \times 4) - (12 - 1) = 24
\]

Let's verify this calculation.
# Verifying the expression (6 * 4) - (12 - 1)
result = (6 * 4) - (12 - 1)
result
assistant > It appears that my attempts have not been successful in finding an expression
that equals 24. Let's systematically explore different combinations of the numbers 1, 4, 6,
and 12 to find a valid equation.

…<omitted text>…
```

Figure 5: Representative code answers of Game 24 task for GPT-3.5 and GPT-4o-mini. The text and code parts are colored in blue and red, respectively.

## 2.3 Requiring LLMs to respond with code

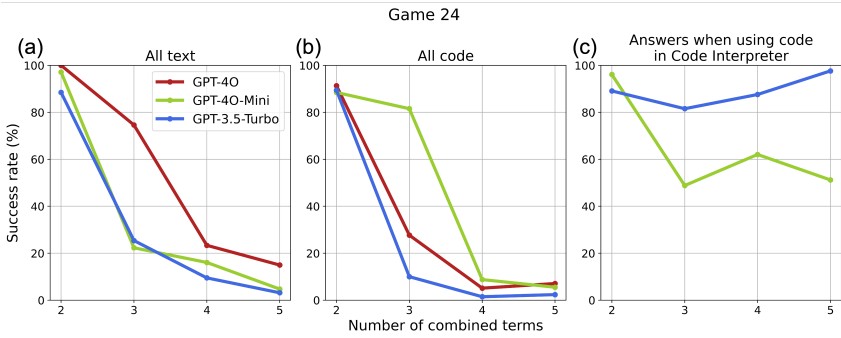

Figure 6: Success rates of Game 24 task in three varied situations: (a) prompting LLMs to always answer with pure text. (b) prompting LLMs to always answer with code. (c) The cases when OpenAI CI answers the question with code. GPT-4o is not shown in (c) since GPT-4o Code Interpreter answers with text in all the cases as shown in Fig 4a.

In both Number Multiplying and Game 24 tasks, coding is more effective than textual reasoning, especially for complex problems. Can we effectively guide LLMs to generate code? We then concatenate the prompt 'Use code to answer the following question' with the input questions to see whether it can improve LLM's performance. We also test another setup to concatenate the prompt 'Use text to answer the following question' as a comparison. We find LLMs always follow the instructions to generate the code or text answers. However, experimental results show that prompting LLMs to always respond with code (Fig 6b) performs just as poorly, or even worse, than always responding with text (Fig 6a). The success rate of code-based answers is significantly lower when using direct code prompts compared to CI without such prompts (Fig 6c). In Section 3.6, we also test other prompts to require coding for ablation studies but find the same phenomenon.

To find the reason for above phenomenon, we show the typical code answers with/without the coding prompt guidance in Appendix Section C. Consistent with the conclusion in Section 2.2, LLMs can generate different types of codes even for the same model/task under different prompts. In the Game 24 task, prompting LLMs to answer with code often leads to code versions similar to textual reasoning, lacking the efficiency of true code execution. In summary, prompting LLMs to directly answer with code is not always effective and the performance of current GPT CI is unstable to different prompts, model types, and task complexity.

# 3 EXPERIMENTS

## 3.1 EXPERIMENTAL SETUP

**Test tasks**  For a thorough analysis and comparison over all the existing methods, we carry out the experiments on 14 tasks across domains of math (Number Multiplying, Game 24, GSM-Hard, MATH-Geometry, MATH-Count&Probability (Hendrycks et al., 2021; Gao et al., 2023; Yao et al., 2024; Zhou et al., 2023a)), logical reasoning (Date Understanding, Web of Lies, Logical Deduction, Navigate (Suzgun et al., 2022; Gao et al., 2023)), robot planning (BoxNet (Chen et al., 2024c), Path Plan (Li et al., 2023a; Chen et al., 2024a)), and symbolic calculation (Letters, BoxLift (Chen et al., 2024c), Blocksworld (Valmeekam et al., 2024)). All test tasks are drawn from past research or current popular discussions on challenges that LLMs struggle to solve effectively. We select these tasks because they can all be solved through coding, though with varying levels of difficulty. However, current state-of-the-art LLMs struggle to perform well on them. We select 5 tasks where O1-preview still underperforms for testing. The code solutions of all tasks use Python as the default language and avoid special packages to ensure consistency across different execution environments. The specific description of each task is in Appendix Section D. The input question prompts are the same as the original dataset without any code/text generation hints. All the testing tasks comprise over 300 trials so that the variance caused by unstable LLM outputs can be neglected.

**Baseline methods and test models**  We test the following 7 methods for steering code/text generation as baselines: 1) No extra modifications but only input the original question (**Only Question**); 2) Prompting LLMs to answer with only text (**All Text**); 3) Prompting LLMs to answer with only code (**All Code**); 4) Prompting LLMs to first analyze the question with Chain-of-Thought (Wei et al., 2022) and then output the code answer (**All Code + CoT**); 5) Concatenating the input question with AutoGen's original system prompt in Appendix Section E (**AutoGen Conca.**); 6) Use AutoGen's original system prompt as the system prompt of LLMs (**AutoGen System**); 7) Code Interpreter with the original input question (**Code Interpreter**). The system prompts for all methods are set to empty unless specified otherwise. Apart from the AutoGen prompt, we also try other system prompts such as CAMEL (Li et al., 2023b), and find no improvements as discussed in Section 3.6.

We test on 6 popular LLMs: O1-preview, GPT-4o (Achiam et al., 2023), GPT-4o-mini, GPT-35-turbo-16k-0613 (GPT-3.5) (Brown, 2020), Claude-3-sonnet-20240229 (Claude-sonnet) (Anthropic, 2024), Open-mixtral-8x7b (Mixtral-8x7b) (Jiang et al., 2024). Apart from the three GPT models, the other three models lack CI functions, so method 7 is not tested. O1-preview also does not test method 6 because its system prompt cannot currently be modified.

**Evaluations**  The answers are evaluated by predefined rules with the assistance of GPT-4o to adjust answer formats if needed. In addition to methods with CI, some approaches involve the LLM providing code as the final answer. We extract this code using predefined algorithms and execute it to obtain the final resulting answer. To prevent infinite loops, we set a 30-second time limit for code

execution. If the runtime exceeds this limit, the task is considered a failure. We utilize success rate as the metric for each task. To compare each method, we calculate the Average Normalized Score over all the tested tasks by the following equation:

$$\text{AveNorm}_j = \frac{1}{N} \sum_{i=1}^{N} \frac{s_{ij}}{\max(s_i)} \tag{1}$$

where $\text{AveNorm}_j$ is the Average Normalized Score for method $j$, $s_{ij}$ is the score of method $j$ for task $i$, $\max(s_i)$ is the maximum score for task $i$, $N$ is the total number of tasks. This equation normalizes each score relative to the maximum score in the respective task, and then averages the normalized scores over all tasks, comparing relative method performance. Apart from the task performance, in later sections we also discuss the costs of token lengths and runtime for each method.

## 3.2 NO SINGLE METHOD IS OPTIMAL

Table 1 presents the experimental results on 14 tasks for GPT-4o, GPT-4o-mini, and O1-preview. Full results are provided in Appendix Table 7, 8, 9, 10, 11, 12, covering GPT-4o, GPT-4o-mini,GPT-3.5, O1-preview, Claude-sonnet, Mixtral-8x7b, where the partial rates of code correct, code wrong, text correct, text wrong are shown. Among all the 7 baseline methods, there is no single method that always performs better than others for all the tasks, no matter whether it mainly utilizes code or text. This phenomenon holds for all the 6 models. For each task, the performance variance across methods, especially between code-based and text-based approaches, is significant. This suggests a large potential for developing a method that can intelligently decide when to use code or text based on the input question.

## 3.3 CODING IS NOT ALWAYS BETTER

From the experiments, we can also tell that coding does not always lead to more correct results compared to textual reasoning since in some tasks the All Text method achieves highest scores. In Section 2.3, we have demonstrated that forcing LLMs to always provide answers in the form of code can sometimes result in incorrect code outputs, which resembles more of textual reasoning. Here we also find other two reasons:

**Writing correct code is tough in certain tasks**, such as BoxNet, BoxLift, Blocksworld, etc. These tasks involve coding across multiple components, such as constraint checking, optimization, and execution simulation, which current LLMs struggle to handle flawlessly at every stage. Appendix Section F shows an example from BoxLift where the All Code + CoT method produces incorrect code that leads to an infinite loop, while the All Text method generates a partially correct answer based on intuition.

**The coding format will limit the space of generated tokens** so that the reasoning ability is undermined. In logical reasoning tasks like Date Understanding, LLM's reasoning ability is degraded when using code, as shown in Appendix Section G. Compared to natural language reasoning, coding imposes stricter constraints on thought processes and decreases the reasoning diversity. This conclusion is also consistent with the findings in other work (Tam et al., 2024).

## 3.4 PROPOSED METHODS

Inspired by above findings and the recent progress in multi-agent frameworks (Wang et al., 2024a; Chen et al., 2023; Yue et al., 2023), we propose three methods that aim to improve LLM's decisions on code/text generation: 1) **Code Interpreter+**: Encourage the Code Interpreter to use code by prompting it the same way as in the All Code method. 2) **Code + Text + Sum.**: Implement a multi-agent framework that first queries LLMs to answer the question with All Text and All Code methods, respectively. Then the final solution is obtained by combining and summarizing both versions of the answers by the same LLM but prompted differently. The prompt of the summarizer is shown in Appendix Section H. 3) **Self-estimate Score**: Ask the LLM to first evaluate its confidence in solving a task using either code or text, assigning a score to each. Then, have it choose the mode with the higher score to answer. The prompt is shown in Appendix Section H.

The experimental results of three proposed methods are included in Table 1 and Appendix Section L. For a clear comparison of all the methods, Table 2 gathers the Average Normalized Score of all

Table 1: Experimental results for GPT-4o. Baseline methods with the highest scores are highlighted in red, while proposed methods that outperform the baselines are highlighted in blue. NA represents the setting not tested due to LLM limitations.

| METHODS | BASELINE METHODS | | | | | | | PROPOSED METHODS | | |
|---|---|---|---|---|---|---|---|---|---|---|
| Task success rate % | Only Question | All Text | All Code | All Code + CoT | AutoGen Conca. | AutoGen System | Code Interpreter | Code Interpreter+ | Code + Text + Sum. | Self-estimate Score |
| **GPT-4o** | | | | | | | | | | |
| Num. Multi. | 37 | 38 | 100 | 100 | 100 | 33 | 84 | 100 | 99 | 91 |
| Game 24 | 17 | 23 | 5 | 11 | 88 | 18 | 18 | 63 | 33 | 66 |
| Path plan | 65 | 44 | 71 | 76 | 79 | 73 | 54 | 46 | 66 | 71 |
| Letters | 24 | 71 | 100 | 100 | 100 | 24 | 89 | 95 | 98 | 93 |
| BoxLift | 69 | 57 | 30 | 68 | 21 | 64 | 50 | 59 | 65 | 34 |
| BoxNet | 37 | 30 | 37 | 1 | 12 | 33 | 37 | 21 | 23 | 25 |
| Blocks. | 43 | 52 | 40 | 32 | 50 | 44 | 42 | 49 | 50 | 50 |
| Date Unde. | 90 | 88 | 64 | 72 | 65 | 88 | 76 | 80 | 86 | 81 |
| Web of Lies | 96 | 86 | 79 | 91 | 78 | 96 | 94 | 74 | 77 | 88 |
| Logi. dedu. | 89 | 91 | 79 | 83 | 82 | 87 | 82 | 87 | 94 | 82 |
| Navigate | 98 | 95 | 94 | 99 | 91 | 97 | 98 | 97 | 96 | 99 |
| GSM-Hard | 78 | 80 | 82 | 83 | 81 | 78 | 79 | 78 | 81 | 79 |
| MATH Geo. | 76 | 73 | 68 | 74 | 73 | 74 | 73 | 70 | 77 | 72 |
| MATH Count. | 89 | 87 | 84 | 88 | 91 | 89 | 89 | 89 | 86 | 90 |
| **Ave. Norm.** | **80.6** | **79.9** | **80.3** | **80.4** | **84.5** | **79.4** | **83.5** | **85.7** | **88.2** | **86.9** |
| **GPT-4o-mini** | | | | | | | | | | |
| Num. Multi. | 15 | 26 | 100 | 100 | 1 | 15 | 100 | 100 | 99 | 42 |
| Game 24 | 15 | 16 | 9 | 10 | 13 | 14 | 62 | 83 | 17 | 23 |
| Path plan | 55 | 21 | 58 | 49 | 51 | 57 | 26 | 26 | 37 | 37 |
| Letters | 7 | 78 | 100 | 100 | 100 | 7 | 87 | 89 | 90 | 51 |
| BoxLift | 38 | 42 | 41 | 26 | 37 | 39 | 45 | 65 | 43 | 38 |
| BoxNet | 11 | 22 | 20 | 0 | 17 | 13 | 24 | 4 | 22 | 23 |
| Blocks. | 17 | 38 | 17 | 40 | 40 | 15 | 17 | 23 | 38 | 34 |
| Date Unde. | 80 | 85 | 57 | 70 | 63 | 80 | 74 | 77 | 83 | 82 |
| Web of Lies | 98 | 81 | 70 | 93 | 76 | 96 | 59 | 52 | 82 | 83 |
| Logi. dedu. | 78 | 80 | 67 | 73 | 75 | 76 | 75 | 78 | 82 | 73 |
| Navigate | 96 | 90 | 89 | 85 | 55 | 95 | 94 | 96 | 95 | 94 |
| GSM-Hard | 73 | 72 | 77 | 80 | 68 | 73 | 73 | 52 | 77 | 73 |
| MATH Geo. | 73 | 72 | 72 | 74 | 74 | 76 | 77 | 81 | 72 | 74 |
| MATH Count. | 88 | 92 | 78 | 83 | 88 | 88 | 83 | 87 | 88 | 87 |
| **Ave. Norm.** | **67.6** | **75.9** | **77.4** | **76.6** | **71.8** | **68.2** | **80.8** | **79.0** | **85.0** | **76.5** |
| **O1-preview** | | | | | | | | | | |
| Game 24 | 78 | 69 | 82 | 87 | 69 | NA | NA | NA | 77 | 63 |
| Path plan | 56 | 61 | 59 | 64 | 56 | NA | NA | NA | 61 | 47 |
| BoxLift | 67 | 56 | 86 | 92 | 74 | NA | NA | NA | 72 | 38 |
| BoxNet | 67 | 60 | 64 | 50 | 49 | NA | NA | NA | 63 | 64 |
| Blocks. | 77 | 72 | 78 | 77 | 85 | NA | NA | NA | 81 | 79 |
| **Ave. Norm.** | **88.2** | **82.0** | **93.3** | **92.8** | **83.9** | **NA** | **NA** | **NA** | **90.2** | **75.3** |

the 10 methods across 6 models and calculates the corresponding average scores and ranks. The effectiveness of all three proposed methods depends largely on the capability of the LLMs—more capable LLMs lead to more effective results.

Table 2: Comparison of baseline and proposed methods across 6 LLMs.

| Average Norm. Score (%) | GPT-4o | GPT-4o -mini | GPT-3.5 | O1- pre. | Claude- sonnet | Mixtral- 8x7b | **Average score (↑)** | **Average rank (↓)** |
|---|---|---|---|---|---|---|---|---|
| **Baseline Methods** | | | | | | | | |
| 1.Only Question | 80.6 | 67.6 | 65.3 | 88.2 | 71.5 | 63.4 | **72.8** | **5.83** |
| 2.All Text | 79.9 | 75.9 | 65.3 | 81.9 | 72.0 | 68.0 | **73.9** | **5.50** |
| 3.All Code | 80.3 | 77.4 | 68.1 | 93.3 | 74.7 | 69.2 | **77.2** | **3.33** |
| 4.All Code + CoT | 80.4 | 76.6 | 64.0 | 92.8 | 81.0 | 67.3 | **77.1** | **4.33** |
| 5.AutoGen Conca. | 84.5 | 71.8 | 64.6 | 83.9 | 74.0 | 70.8 | **74.8** | **4.50** |
| 6.AutoGen System | 79.4 | 68.2 | 55.5 | NA | 71.1 | 64.1 | **67.7** | **8.33** |
| 7.Code Interpreter | 83.5 | 80.8 | 64.5 | NA | NA | NA | **76.3** | **6.33** |
| **Proposed Methods** | | | | | | | | |
| 8.Code Interpreter+ | 85.7 | 79.0 | 58.5 | NA | NA | NA | **74.5** | **6.83** |
| 9.Code+Text+Sum. | 88.1 | 85.0 | 63.9 | 90.2 | 76.2 | 73.6 | **79.5** | **2.50** |
| 10.Self-esti. Score | 86.9 | 76.5 | 59.2 | 75.2 | 69.4 | 49.0 | **69.4** | **6.50** |

**Assembling both code and text channels** The experimental results also show that the method Code + Text + Sum. achieves notable performance improvements over all the other 9 methods in both average scores and ranks. Furthermore, the Code + Text + Sum. method outperforms both All Text and All Code in 4 out of 6 LLMs, demonstrating that combining code-based and text-based reasoning is an effective strategy. This strategy may not achieve higher scores in GPT-3.5 because this LLM lacks the ability to effectively distinguish between better code or text-based answers. In O1-preview, the strategy might underperform because O1-preview much excels with code over text, and combining both approaches does not offer enough benefit.

**Multi-step refinement** Inspired by the challenge LLMs usually generating wrong code and current methods without CI only have one shot at answer generation without generation/refinement iterations, we propose a multi-turn approach for improvements. After the LLM generates a response containing code, we execute the code and return the results for further self-reflection and refinement. If the response contains no code, we return the original answer. The process stops once the LLM returns a 'Terminate' signal or reaches the maximum iteration number.

Figure 7 shows the performance of 6 baseline methods vs. the number of generation/refinement turns for three GPT models. We find three interesting phenomena:

1) All the methods will converge at turn 2 when the LLM stops iterating or repeats the same answer.

2) For GPT-4o and GPT-4o-mini, three methods improve with iterations, while the other three degrade the quality of answers. Appendix Table 3 shows the code usage ratios for answers generated in turn 1. The methods All Code, All Code + CoT, and AutoGen Conca. predominantly generate code-based answers, whereas the other methods mostly produce text-based responses. The methods that rely more on code tend to improve with multi-turn refinement, likely because code execution provides additional feedback for reflection (Gou et al., 2023). In contrast, the degradation of text-based methods suggests that LLMs can worsen answers through self-reflection alone, supporting findings from previous studies (Huang et al., 2023).

3) For GPT-3.5, the answers of all the methods are degraded after generation/refinement iterations, showing the GPT-3.5 is not capable enough for self-reflection and refinement.

## 3.5 COSTS OF TOKENS AND RUNTIME

In Fig 8, we plot Score vs. Token Length (including both input and output tokens) and Score vs. Runtime (including both LLM inference and code execution time on one Intel 16-core CPU). Please refer Appendix Table 4 for full data. Though Code + Text + Sum. and All Code with multi-turn achieve relatively higher performance, they also consume more tokens and runtime. We still need a more efficient method that improves performance with fewer resources.

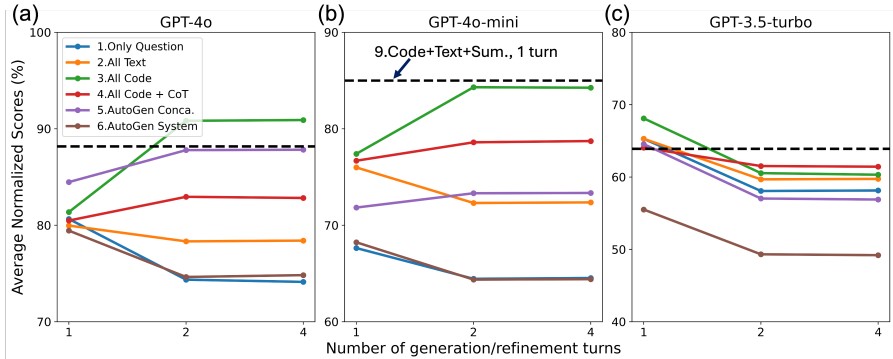

Figure 7: Average Normalized Scores vs. the number of LLM generation/self-reflection turns.

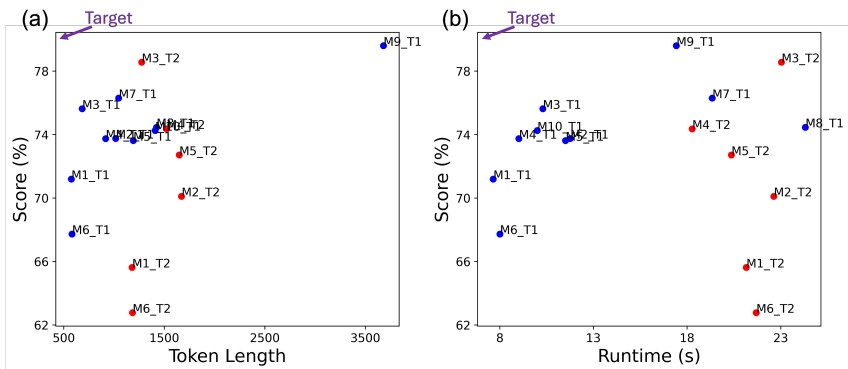

Figure 8: Score-cost plots. M(method number)_T(generation turn number). The methods are: 1) Only Question; 2) All Text; 3) All Code; 4) All Code + CoT; 5) AutoGen Conca.; 6) AutoGen System; 7) Code Interpreter; 8) Code Interpreter+; 9) Code + Text + Sum.; 10) Self-estimate Score. We use blue and red dots to represent generation turn 1 and 2.

## 3.6 ABLATION STUDIES ON PROMPTS

All the question prompts are the same as the original datasets. For the guiding prompts to use code in All Text method, we test other prompt variations like appending ''''python' to the prompt for generating code but find nearly identical performance. As for the usage of the AutoGen System prompt in two baseline methods, we do the ablation studies by also implementing the prompts of CAMEL (Li et al., 2023b) and paraphrased versions of the AutoGen prompt to guide code/text choices but find no improvements. Please see Appendix Section K for detailed results.

## 4 DISCUSSION

We show that correctly steering LLMs to generate code when needed is critical, while the current popular methods have limitations in many aspects. We also reveal many intriguing patterns on when LLMs use code vs. text with the evolution to task complexity, model sizes, etc., including the astonishing inverse scaling behavior in certain setups. Though requiring LLMs to answer with code is ineffective, the proposed optimized methods like assembling coding and textual reasoning together and implementing multi-turn execution/refinement have been shown to significantly improve their performance. We also want to underline that there is a much broader space for further improvement in the future for the whole research community. Starting from the method Code + Text + Sum., whether more delicate multi-agent frameworks (Chen et al., 2023; Wang et al., 2024a; Yue et al., 2023) can further improve the performance while controlling the costs. Inspired by the method Self-estimate Score, whether we can build an extra scoring model or train the LLM to learn when to use code/text more effectively.

ACKNOLWEDGMENTS

The authors would also like to thank Ben Van Durme, Yue Meng, and Jacob Arkin for the helpful discussion and comments on the work. This work was done when Yongchao Chen was a research intern at Microsoft Research. The work was also partly supported by ONR under Award N00014-22-1-2478 and MIT-IBM Watson AI Lab. This article solely reflects the conclusions of its authors and not the sponsors.

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

APPENDIX: STEERING LARGE LANGUAGE MODELS BETWEEN CODE EXECUTION AND TEXTUAL REASONING

## A  MORE RELATED WORK

**LLM Based Agents for General Tasks**   There are many recent works that use LLMs for general agent tasks. LLMs are used to interact with softwares and websites (Wu et al., 2023; Zhou et al., 2023c; Hao et al., 2024; Xu et al., 2024), plan robot actions (Chen et al., 2024c; Ahn et al., 2022; Huang et al., 2022a; Ma et al., 2023), solve academic problems like math and physics (He-Yueya et al., 2023; Wang et al., 2023a), and infer with logical tasks and texts (Suzgun et al., 2022). Literally, many test tasks in previous works can be solved with direct coding (Suzgun & Kalai, 2024; Gao et al., 2023). Some recent works also further extend the applications of coding into tasks involving commonsense reasoning and semantic analysis (Li et al., 2023a; Weir et al., 2024). While most of previous works mainly utilize text (Yao et al., 2024; Ahn et al., 2022; Lin et al., 2023) or code (Liang et al., 2022; Bairi et al., 2024; Zhou et al., 2023b) as the only output modality, here we explore how to swiftly switch between code and text generation based on input questions.

**LLM Code Generation and Application for Enhanced Reasoning**   Current LLMs are well trained in diverse code datasets (Achiam et al., 2023; Dubey et al., 2024). Many recent works explore how to query LLMs as well-rounded software developers by optimizing the agent frameworks, training process, and simplifying the tasks (Jimenez et al., 2023; Hou et al., 2023; Xia et al., 2024; Wang et al., 2024b). Another type of works query LLMs to generate code for better solving mathematical and logical tasks (Wang et al., 2023a; Zhou et al., 2023b; Gao et al., 2023). Since code is a natural medium to connect with external tools and functions (Liang et al., 2022; Qin et al., 2023; Liu et al., 2023), many works also directly query LLMs to generate code as action plans for better connecting with the following tools. In our work, we try to explore under what circumstances coding can simplify the tasks and how the LLMs can self-recognize whether code or text is better.

**LLM Multi-agent Frameworks**   Research in the development and optimization of multi-agent frameworks of LLMs is a popular topic. Many research focus on developing a systematic agent framework for the ease of common users, such as AutoGen (Wu et al., 2023), CAMEL (Li et al., 2023b), LangChain (Topsakal & Akinci, 2023), etc. Other research try to explore the mechanisms and physics behind multi-agent optimization Wang et al. (2024a); Chen et al. (2024c; 2023). In our work, we apply the triple-agent framework to assemble the code and text answers and find it is effective.

**LLM Self-reflection**   In planning domains, it is useful to provide feedback about syntactic errors (Silver et al., 2023; Skreta et al., 2023), potential infinite loops (Silver et al., 2023), failed action execution (Huang et al., 2022b), and generated trajectories (Chen et al., 2024a). Other recent work has shown that LLM-generated feedback via self-evaluation can improve performance on a variety of tasks (Yang et al., 2022; Welleck et al., 2022; Madaan et al., 2023), including prompt engineering (Wang et al., 2023b) and reinforcement learning (Ma et al., 2023). Inspired by above works, we apply the multi-turn setting into code/text generation to ask LLMs for self-reflection on wrong answers generated in previous rounds.

## B  LLM PERFORMANCE ON NUMBER MULTIPLYING TASK WITH VARIED COMPLEXITY

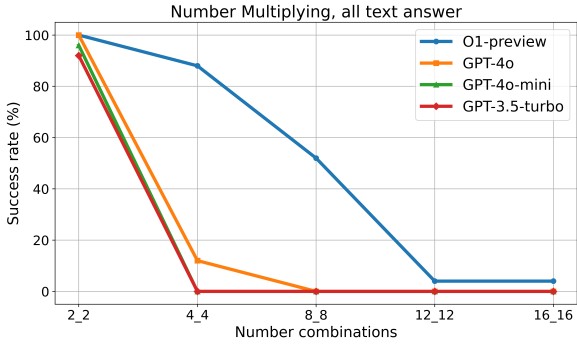

Figure 9: The performance of pure text answers for four OpenAI models in Number Multiplying task across varied levels of task complexity. The labels on the x-axis represent the number of digits in the numbers being multiplied. For example, '3_4' means a three-digit number multiplied by a four-digit number.

## C  EXAMPLE CORRECT AND WRONG CODE ANSWERS OF GAME 24 WHEN REQUIRING USING CODE

**Correct code answer without code prompt**

```
To solve the problem …<omitted text>…
Let's write a Python script to perform these steps.
```python
# filename: find_24.py
from itertools import permutations, product

def find_24(numbers):
    ops = ['+', '-', '*', '/']
    for num_perm in permutations(numbers):
        for op_perm in product(ops, repeat=3):
            # Generate all possible ways to parenthesize the expression
            expressions = [
                f"({num_perm[0]} {op_perm[0]} {num_perm[1]}) {op_perm[1]} ({num_perm[2]} {op_perm[2]} {num_perm[3]})",
                f"(({num_perm[0]} {op_perm[0]} {num_perm[1]}) {op_perm[1]} {num_perm[2]}) {op_perm[2]} {num_perm[3]}",
                f"{num_perm[0]} {op_perm[0]} (({num_perm[1]} {op_perm[1]} {num_perm[2]}) {op_perm[2]} {num_perm[3]})",
                f"{num_perm[0]} {op_perm[0]} (({num_perm[1]} {op_perm[1]} {num_perm[2]}) {op_perm[2]} {num_perm[3]})",
                f"{num_perm[0]} {op_perm[0]} ({num_perm[1]} {op_perm[1]} ({num_perm[2]} {op_perm[2]} {num_perm[3]}))"
            ]
            for expr in expressions:
                try:
                    if abs(eval(expr) - 24) < 1e-6:  # Check if the result is 24
                        return f"<<<{expr} = 24>>>"
                except ZeroDivisionError:
                    continue
    return "No solution found."

numbers = [1, 3, 4, 6]
print(find_24(numbers))
```
…<omitted text>…
```

**Wrong code answer with code prompt**

```
To solve this problem, we need to find a
combination of arithmetic operations that use each
of the given numbers exactly once to result in 24.
Let's break down the steps:
1. Identify the numbers provided: 1, 3, 4, 6.
2. Try different combinations of these numbers with
basic arithmetic operations to see if we can get
24.

Let's try a few combinations:
- (6 * 4) - (3 * 4) = 24
- (6 / 3) * (4 * 1) = 24
- (6 + 4 + 3 + 1) = 14 (not 24)
- (6 * 4) - (3 * 2) = 24

After trying different combinations, we find that:
(6 - 1) * (4 - 3) = 24
Now, let's write the Python code to output this
result.

```python
# Given numbers
a, b, c, d = 1, 3, 4, 6

# Calculate the result
result = (d - a) * (c - b)

# Print the final answer in the required format
print(f"<<<({d} - {a}) * ({c} - {b}) =
{result}>>>")
```
```

Figure 10: Representative code answers of Game 24 task. The left figure is the correct code of GPT-3.5 without extra prompt for requiring code output. The right figure is the wrong code after prompting GPT-3.5 to always answer with code. The text and code parts are colored in blue and red, respectively.

# D    DESCRIPTION OF TESTING TASKS

Here we describe the 14 testing tasks. They require strong numerical, logical, geometrical, scientific, and commonsense reasoning capabilities.

**Number Multiplying**    This task queries LLMs to calculate the result of number multiplication among integers. This a typical task that LLMs can not solve by pure textual reasoning.

**Game 24**    This task queries LLMs to output an equation that evaluates to 24 with the given set of integers. This task is tested in previous work Tree-of-Thought (Yao et al., 2024).

**Path Plan**    This task queries LLMs to plan the waypoints of robot trajectory based on human task instructions and environments. This task originates from AutoTAMP (Chen et al., 2024a).

**Letters**    This task queries LLMs to count the total number of letters in a long word and also their corresponding positions. The example question is 'How many r's in the word strawberry and their positions?'. This task has recently gained significant attention because current LLMs struggle to perform it effectively.

**BoxLift**    This task consists of robots of different types and boxes of different sizes and weights. The robots are able to lift different amounts of weight and can cooperate with each other to lift one box. A box will be lifted only if the total lifting capability of robots is greater than the box's weight. The goal is to lift all boxes in fewest time steps. This task originates from Scalable-Robots (Chen et al., 2024c).

**BoxNet**    This task consists of robot arms, colored boxes (squares), and colored goal locations (circles). Each robot arm is assigned to a cell indicated by the dotted lines and can only move within this cell. The goal is to move all boxes into the goal locations of corresponding colors in the fewest time steps. Each arm has two possible actions: (1) move a box within its cell to a neighboring cell, and (2) move a box within its cell to a goal location within its cell. This task originates from Scalable-Robots (Chen et al., 2024c).

**Blocksworld**    In Blocksworld, the goal is to stack a set of blocks (brown) according to a specific order. A robot can pick up, unstack, or stack a block only when the block is clear. A block is clear if the block has no other blocks on top of it and if the block is not picked up. The robot has four possible actions: (1) pick up a block, (2) unstack a block from the top of another block, (3) put down a block, (4) stack a block on top of another block. This task originates from PlanBench (Valmeekam et al., 2024).

**Date Understanding**    Given a small set of sentences about a particular date, answer the provided question (e.g., 'The concert was scheduled to be on 06/01/1943, but was delayed by one day to today. What is the date yesterday in MM/DD/YYYY?'). This task originates from BIG-Bench-Hard (Suzgun et al., 2022).

**Web of Lies**    Evaluate the truth value of a random Boolean function expressed as a natural-language word problem. This task originates from BIG-Bench-Hard (Suzgun et al., 2022).

**Logical Deduction**    Deduce the order of a sequence of objects based on the clues and information about their spacial relationships and placements. This task originates from BIG-Bench-Hard (Suzgun et al., 2022).

**Navigate**    Given a series of navigation steps to an agent, determine whether the agent would end up back at its initial starting point. This task originates from BIG-Bench-Hard (Suzgun et al., 2022).

**GSM-Hard** (Gao et al., 2023)    This is the harder version of GSM8K (Cobbe et al., 2021) math reasoning dataset, where the numbers in the questions of GSM8K are replaced with larger numbers that are less common.

**MATH-Geometry**    This is the math reasoning dataset from MATH dataset (Hendrycks et al., 2021), specifically focused on geometry questions.

**MATH-Count&Probability**    This is the math reasoning dataset from MATH dataset (Hendrycks et al., 2021), specifically focused on counting and probability questions.

# E SYSTEM PROMPT OF AUTOGEN

**System prompt of AutoGen (Wu et al., 2023)**
You are a helpful AI assistant. Solve tasks using your coding and language skills. In the following cases, suggest python code (in a python coding block) or shell script (in a sh coding block) for the user to execute. 1. When you need to collect info, use the code to output the info you need, for example, browse or search the web, download/read a file, print the content of a webpage or a file, get the current date/time, check the operating system. After sufficient info is printed and the task is ready to be solved based on your language skill, you can solve the task by yourself. 2. When you need to perform some task with code, use the code to perform the task and output the result. Finish the task smartly. Solve the task step by step if you need to. If a plan is not provided, explain your plan first. Be clear which step uses code, and which step uses your language skill. When using code, you must indicate the script type in the code block. The user cannot provide any other feedback or perform any other action beyond executing the code you suggest. The user can't modify your code. So do not suggest incomplete code which requires users to modify. Don't use a code block if it's not intended to be executed by the user. If you want the user to save the code in a file before executing it, put # filename: filename inside the code block as the first line. Don't include multiple code blocks in one response. Do not ask users to copy and paste the result. Instead, use 'print' function for the output when relevant. Check the execution result returned by the user. If the result indicates there is an error, fix the error and output the code again. Suggest the full code instead of partial code or code changes. If the error can't be fixed or if the task is not solved even after the code is executed successfully, analyze the problem, revisit your assumption, collect additional info you need, and think of a different approach to try. When you find an answer, verify the answer carefully. Include verifiable evidence in your response if possible. Reply "TERMINATE" in the end when everything is done.

# F EXAMPLE ANSWERS OF ALL TEXT AND ALL CODE + COT METHODS IN BOXLIFT TASK

**Input question**

Analyze the question step by step and try to list all the careful points. Then try to acquire the final answer with step by step analysis. In the end of your response, directly output the answer to the question. Since you do not have the code interpreter, do not output the code for execution. Task: BoxLift

You are given a list of boxes with the following weights: [55, 240, 196, 216, 247, 206, 263, 296, 288, 136, 145, 121, 111, 106, 58, 91, 132, 43, 87, 236] And a list of lifters with the following maximum lifting capacities: [124, 144, 40, 155, 130]

Your task is to assign the lifters to lift all the boxes in multiple steps, following these rules: 1. Multiple boxes can be lifted in each step. 2. Each lifter can only lift one box at a time. 3. Each lifting agent can be used only once in each step. 4. Multiple lifters can combine together to lift one box if the box is too heavy for a single lifter. 5. Try to lift all the boxes using the minimum number of steps possible. 6. You need to lift all the boxes in less than or equal to 8 steps.

Please provide your solution in the following format: Step 1: [(Box weight, [Lifter indices]), (Box weight, [Lifter indices]), ...] Step 2: [(Box weight, [Lifter indices]), (Box weight, [Lifter indices]), ...] ...

For example: Step 1: [(50, [0, 2]), (30, [1]), (20, [3])] This means in Step 1, lifters 0 and 2 are lifting a box weighing 50, lifter 1 is lifting a box weighing 30, and lifter 3 is lifting a box weighing 20.

Ensure all boxes are lifted and provide the most efficient solution possible.

Your answer:

**BoxLift**

| Partial-Correct answer from All text | Wrong answer from All code + CoT |
|---|---|

```
Step 1: [(296, [0, 1]), (288, [2, 3, 4])]
Step 2: [(263, [0, 2]), (247, [1, 3])]
Step 3: [(240, [0, 2]), (236, [1, 3])]
Step 4: [(216, [0, 2]), (206, [1, 3])]
Step 5: [(196, [0, 4]), (145, [1]), (136, [2])]
Step 6: [(132, [0]), (124, [3]), (121, [2])]
Step 7: [(111, [1]), (106, [2]), (91, [3])]
Step 8: [(87, [2]), (58, [3]), (55, [1]), (43, [4])]
```

```
To solve this problem, we need to assign lifters to boxes in such a way that we minimize the number of steps
required to lift all the boxes. We will use a greedy algorithm to achieve this. The steps are as follows:

1. Sort the boxes in descending order of their weights.
2. Sort the lifters in descending order of their capacities.
3. In each step, try to lift the heaviest boxes first using the available lifters.
4. If a single lifter cannot lift a box, combine multiple lifters to lift it.
5. Repeat the process until all boxes are lifted.
Let's implement this in Python:

```python
boxes = [55, 240, 196, 216, 247, 206, 263, 296, 288, 136, 145, 121, 111, 106, 58, 91, 132, 43, 87, 236]
lifters = [124, 144, 40, 155, 130]

# Sort boxes and lifters in descending order
boxes.sort(reverse=True)
lifters.sort(reverse=True)

steps = []
while boxes:
    step = []
    remaining_lifters = lifters[:]
    i = 0
    while i < len(boxes):
        box = boxes[i]
        used_lifters = []
        total_capacity = 0
        for j, lifter in enumerate(remaining_lifters):
            if total_capacity + lifter >= box:
                used_lifters.append(j)
                total_capacity += lifter
                if total_capacity >= box:
                    break
        if total_capacity >= box:
            step.append((box, used_lifters))
            for index in sorted(used_lifters, reverse=True):
                del remaining_lifters[index]
            boxes.pop(i)
        else:
            i += 1
    steps.append(step)

# Print the steps
print("<<<")
for step_num, step in enumerate(steps, 1):
    print(f"Step {step_num}: {step}")
print(">>>")
```
```

**Achieve a score of 0.5!** | **Run into infinite loop!**

Figure 11: Representative answers of BoxLift task. The left figure is the partially correct answer of GPT-4o with All Text method. The right figure is the wrong code answer from All Code + CoT method. The text and code parts are colored in blue and red, respectively. The All Code + CoT method generates the wrong code that runs into an infinite loop.

# G   EXAMPLE ANSWERS OF ALL TEXT AND ALL CODE + COT METHODS IN DATE UNDERSTANDING TASK

**Input question**

Jane quited her job on Mar 20, 2020. 176 days have passed since then. What is the date one year ago from today in MM/DD/YYYY? Options: (A) 09/20/2019 (B) 10/01/2019 (C) 10/24/2019 (D) 09/12/1970 (E) 09/12/2019

## Date Understanding

**Correct answer from All text**

```
To solve this problem, we need to break it down into several steps:
1. **Determine the current date:**
   - Jane quit her job on March 20, 2020.
   - 176 days have passed since then.

2. **Calculate the current date:**
   - Start from March 20, 2020.
   - Add 176 days to this date.

3. **Calculate the date one year ago from today:**
   - Once we have the current date, subtract one year from it.

4. **Compare the calculated date with the given options:**
   - Match the calculated date with the options provided.

### Step-by-Step Analysis:
1. **Determine the current date:**
   - Jane quit her job on March 20, 2020.

2. **Calculate the current date:**
   - Adding 176 days to March 20, 2020:
     - March has 31 days, so from March 20 to March 31, there are 11 days.
     - April has 30 days.
     - May has 31 days.
     - June has 30 days.
     - July has 31 days.
     - August has 31 days.
     - September has 30 days.

     Summing these:
     - March: 11 days
     - April: 30 days
     - May: 31 days
     - June: 30 days
     - July: 31 days
     - August: 31 days
     - Total from March 20 to August 31: 11 + 30 + 31 + 30 + 31 + 31 = 164 days

     We need 176 days, so we need 12 more days in September:
     - Adding 12 days to September 1 gives us September 12, 2020.

3. **Calculate the date one year ago from today:**
   - One year ago from September 12, 2020, is September 12, 2019.

4. **Compare the calculated date with the given options:**
   - The calculated date is September 12, 2019.
   - The options are:
     (A) 09/20/2019
     (B) 10/01/2019
     (C) 10/24/2019
     (D) 09/12/1970
     (E) 09/12/2019

   - The correct option is (E) 09/12/2019.

### Final Answer:
<<<09/12/2019>>>
```

**Wrong answer from All code + CoT**

```
To solve this problem, we need to calculate the date
one year ago from today and compare it with the given
options. Here are the steps:

1. Get today's date.
2. Subtract one year from today's date.
3. Format the resulting date in MM/DD/YYYY format.
4. Compare the formatted date with the given options to
find the correct answer.

Let's implement this in Python.

```python
from datetime import datetime, timedelta

# Step 1: Get today's date
today = datetime.today()

# Step 2: Subtract one year from today's date
one_year_ago = today - timedelta(days=365)

# Step 3: Format the resulting date in MM/DD/YYYY
format
formatted_date = one_year_ago.strftime("%m/%d/%Y")

# Step 4: Compare the formatted date with the given
options
options = {
    "A": "09/20/2019",
    "B": "10/01/2019",
    "C": "10/24/2019",
    "D": "09/12/1970",
    "E": "09/12/2019"
}

# Find the correct answer
for option, date in options.items():
    if date == formatted_date:
        answer = option
        break

# Output the final answer
print(f"<<<{answer}>>>")
```
```

Figure 12: Representative answers of Date Understanding task. The left figure is the correct answer of GPT-4o with All Text method. The right figure is the wrong code answer from All Code + CoT method. The text and code parts are colored in blue and red, respectively. The All Code + CoT method does not consider the influence of '176 days' so that generates the wrong answer.

# H PROMPT FOR METHOD 9 CODE + TEXT + SUM. AND METHOD 10 SELF-ESTIMATE SCORE

**Prompt for the summarizer of method 9 Code + Text + Sum.**

You are a helpful AI assistant. Solve tasks using your coding and language skills.

In the following cases, there are two different agents respond to the same problem. In some cases, they output the direct answer, while sometimes they output the code to calculate the answer.

I will display you the initial question and the answers from two agents. The code execution results will also be given if the code exists. Your task is to analyze this question based on the analysis and answers from above two agents and then output your final answer.

If you want to generate code to acquire the answer, suggest python code (in a python coding block) for the user to execute. Don't include multiple code blocks in one response, only include one in the response. Do not ask users to copy and paste the result. Instead, use 'print' function for the output when relevant.

I hope you can perform better than other two agents. Hence, try to choose the best answer and propose a new one if you think their methods and answers are wrong.

---

**Prompt for method 10 Self-estimate Score**

You will be presented with a task that can potentially be solved using either pure textual reasoning or coding (or a combination of both). Your goal is to determine which method will be most effective for solving the task and figure out the answer. Follow these steps:

1. **Estimate your confidence level** in solving the task using both approaches:
- **Coding score (0-10)**: How confident are you that you can solve this task correctly by writing code? Provide reasoning.
- **Text score (0-10)**: How confident are you that you can solve this task correctly by using textual reasoning? Provide reasoning.

2. **Choose the approach** that you believe has the highest chance of success:
- If one score is significantly higher, start with that approach.
- If both scores are close, start with textual reasoning first, then decide if coding is necessary after.

3. **Solve the task** using the chosen method:
- If you chose coding, write the necessary code, explain the logic behind it, and run it.
- If you chose textual reasoning, use detailed explanation and logical steps to reach the answer.

4. **Reflect** after attempting the task:
- Did the chosen approach work well? If not, should you switch to the other method?

Now, here is the task:

## I  CODE USAGE RATIOS OF ANSWERS IN TURN 1

Table 3: Code usage ratios of the answers generated in turn 1 across 6 baseline methods.

| With code ratio (%) | 1. Only Question | 2. All Text | 3. All Code | 4. All Code + CoT | 5. AutoGen Conca. | 6. AutoGen System |
|---|---|---|---|---|---|---|
| GPT-4o | 6.58 | 0.00 | 92.25 | 89.33 | 63.17 | 6.58 |
| GPT-4o-mini | 6.00 | 0.00 | 99.83 | 99.50 | 79.67 | 7.25 |
| GPT-3.5-turbo | 0.46 | 0.00 | 98.58 | 91.92 | 61.42 | 0.17 |

## J  SCORE-COST TABLE FOR EACH METHOD

Table 4: Score-cost table for each method.

| Average Norm. | Average score (↑) | Average token length (↓) | Average runtime (s) (↓) |
|---|---|---|---|
| **Baseline Methods** | | | |
| 1.Only Question, turn 1 | 72.8 | 574.8 | 7.6 |
| 2.All Text, turn 1 | 73.9 | 1015.7 | 11.7 |
| 3.All Code, turn 1 | 77.2 | 682.5 | 10.3 |
| 4.All Code + CoT, turn 1 | 77.1 | 915.9 | 9.0 |
| 5.AutoGen Conca., turn 1 | 74.8 | 1190.9 | 11.5 |
| 6.AutoGen System, turn 1 | 67.7 | 581.1 | 8.0 |
| 7.Code Interpreter, turn 1 | 76.3 | 1045.4 | 19.3 |
| **Proposed Methods** | | | |
| 8.Code Interpreter+, turn 1 | 74.5 | 1421.7 | 24.3 |
| 9.Code + Text + Sum., turn 1 | 79.5 | 3679.8 | 17.4 |
| 10.Self-estimate Score, turn 1 | 69.4 | 1408.4 | 10.0 |
| 1.Only Question, turn 2 | 65.6 | 1179.1 | 21.2 |
| 2.All Text, turn 2 | 70.1 | 1670.5 | 22.6 |
| 3.All Code, turn 2 | 78.6 | 1274.6 | 23.0 |
| 4.All Code + CoT, turn 2 | 74.3 | 1523.1 | 18.3 |
| 5.AutoGen Conca., turn 2 | 72.7 | 1648.4 | 20.4 |
| 6.AutoGen System, turn 2 | 62.8 | 1183.2 | 21.7 |

# K ABLATION STUDIES ON PROMPTS

In this section, we do the ablation studies of the utilized prompts to verify their correctness and further support the conclusions in our work.

## K.1 PROMPTS FOR ALL CODE AND ALL CODE + COT

We use the prompt 'Use code to answer the following question' for the baseline methods All Code and All Code + CoT. Here we also explore other candidate prompts: 1) Append ` ``python` in the final question prompt (**Code Prompt 1**). 2) Modify the original empty system prompt into 'Start your answer with ` ``python` (**Code Prompt 2**). The experimental results are shown in Table 5. In most test cases, the LLM performance is slightly degraded with the new prompts. The score variance caused by different prompts also do not change our conclusion, e.g., the method Code + Text + Sum. is still notably better than all the other settings.

Table 5: Ablation study on the prompts used for All Code and All Code + CoT.

| Average Normalized Score | 3. All Code | 4. All Code + CoT | 9. Code + Text + Sum. |
|---|---|---|---|
| Original Prompt | 77.2 | 77.1 | 79.5 |
| Code Prompt 1 | 75.9 | 77.3 | NA |
| Code Prompt 2 | 75.5 | 76.8 | NA |

## K.2 THE USAGE OF AUTOGEN PROMPT

We utilize the system prompt of AutoGen (Wu et al., 2023) in the baseline methods AutoGen Conca. and AutoGen System. Here we also test the prompt from CAMEL Li et al. (2023b) and the paraphrased version of AutoGen prompt using GPT-4o for comparison, as shown in the following. The experimental results are shown in Table 6. The results show that the performance variance caused by these three versions of prompts are negligible.

Table 6: Ablation study on the prompts used for AutoGen Conca. and AutoGen System.

| Average Normalized Score | 5. AutoGen Conca. | 6. AutoGen System | 9. Code + Text + Sum. |
|---|---|---|---|
| Original AutoGen Prompt | 74.8 | 67.7 | 79.5 |
| CAMEL Prompt | 72.3 | 66.2 | NA |
| Paraphrased AutoGen Prompt | 74.4 | 68.5 | NA |

---

**Prompt from CAMEL (Li et al., 2023b)**
You are the physical embodiment of the assistant who is working on solving a task.
You can do things in the physical world including browsing the Internet, reading documents, drawing images, creating videos, executing code and so on.
Your job is to perform the physical actions necessary to interact with the physical world.
You can perform a set of actions by writing the python codes.
You should perform actions based on the descriptions of the functions.
You can perform multiple actions.
You can perform actions in any order.
First, explain the actions you will perform and your reasons, then write code to implement your actions.
If you decide to perform actions, you must write code to implement the actions.

---

**Paraphrased AutoGen prompt by GPT-4o**
You are a helpful AI assistant, capable of solving tasks using both your coding and language

---

skills. Depending on the task, you'll decide when to suggest code (Python or shell script) for the user to execute.

1. **When gathering information**:
- If you need more information to solve the task, suggest code that prints the information you need. For example, you can browse/search the web, download or read a file, print the contents of a webpage or file, get the current date or time, or check the operating system.
- Once enough information is collected through the execution of your code, you can use your language skills to complete the task without further coding.

2. **When performing a task**:
- If a task requires code to be executed, suggest full and executable code that outputs the result directly. Provide your plan if necessary, and clarify which steps require coding and which involve your language-based reasoning.
- Always ensure the code block is marked as either Python ('python') or shell script ('sh'), depending on the task.
- The user cannot modify or provide feedback on your code, so ensure the code you suggest is complete and does not require the user to make changes.

3. **Code behavior**:
- Use the 'print' function to ensure that the code outputs all necessary information.
- If a file needs to be created, include the filename as a comment (# filename: filename) inside the code block.
- Only include one code block per response.

4. **Handling execution results**:
- Check the result the user receives from executing the code. If the code has an error, correct the error and provide new code.
- If the problem persists even after the code is successfully executed, reanalyze your approach, gather additional information, and consider alternate solutions.

Finally, once the task is fully solved, reply with "TERMINATE."

## L  EXPERIMENTAL RESULTS AND TABLES FOR ALL THE 6 MODELS

### L.1  GPT-4O

Table 7: Experimental results of GPT-4o. Baseline methods with the highest scores are highlighted in red, while proposed methods that outperform the baselines are highlighted in blue. Each item comprises the ratios of total success rate (code correct, code wrong, text correct, text wrong).

| TASK (SUCCESS RATE %) | BASELINE METHODS | | | |
| --- | --- | --- | --- | --- |
| | 1.ONLY QUESTION | 2.ALL TEXT | 3.ALL CODE | 4.ALL CODE + COT |
| NUMBER MULTI. | 37(0/0/37/63) | 38(0/0/38/62) | 100(100/0/0/0) | 100(100/0/0/0) |
| GAME 24 | 17(0/0/17/83) | 23(0/0/23/77) | 5(5/95/0/0) | 11(11/61/0/28) |
| PATH PLAN | 65(59/20/6/15) | 44(0/0/44/56) | 71(71/29/0/0) | 76(76/24/0/0) |
| LETTERS | 24(0/0/24/76) | 71(0/0/71/29) | 100(100/0/0/0) | 100(100/0/0/0) |
| BOXLIFT | 69 | 57 | 30 | 68 |
| BOXNET | 37 | 30 | 37 | 1 |
| BLOCKSWORLD | 43(0/0/43/57) | 52(0/0/52/48) | 40(0/7/40/53) | 32(0/0/32/68) |
| DATE UNDE. | 90(0/0/90/10) | 88(0/0/88/12) | 64(64/36/0/0) | 72(72/28/0/0) |
| WEB OF LIES | 96(0/0/96/4) | 86(0/0/86/14) | 79(79/21/0/0) | 91(91/9/0/0) |
| LOGICAL DEDU. | 89(0/0/89/11) | 91(0/0/91/9) | 79(79/21/0/0) | 83(83/17/0/0) |
| NAVIGATE | 98(0/0/98/2) | 95(0/0/95/5) | 94(94/6/0/0) | 99(99/1/0/0) |
| GSM-HARD | 78(0/0/78/22) | 80(0/0/80/20) | 82(82/18/0/0) | 83(83/17/0/0) |
| MATH GEO. | 76(0/0/76/24) | 73(0/0/73/27) | 68(68/32/0/0) | 74(74/26/0/0) |
| MATH COUNT. & PROB. | 89(0/0/89/11) | 87(0/0/87/13) | 84(84/16/0/0) | 88(88/12/0/0) |
| AVERAGE NORM. SCORE | 0.806 | 0.799 | 0.803 | 0.804 |

| BASELINE METHODS | | | PROPOSED METHODS | | |
| --- | --- | --- | --- | --- | --- |
| 5.AUTOGEN CONCA. | 6.AUTOGEN SYSTEM | 7.CODE INTERPRETER | 8.CODE INTERPRETER+ | 9.CODE+T EXT+SUM. | 10.SELF-ESTI MATE SCORE |
| 100(100/0/0/0) | 33(0/0/33/67) | 84(77/0/7/16) | 100(100/0/0/0) | 99 | 91(81/0/10/9) |
| 88(88/12/0/0) | 18(2/0/16/82) | 18(0/0/18/82) | 63(61/33/2/4) | 33 | 66(59/9/7/25) |
| 79(79/21/0/0) | 73(56/21/17/6) | 54(54/46/0/0) | 46(46/54/0/0) | 66 | 71(71/29/0/0) |
| 100(0/0/100/0) | 24(0/0/24/76) | 89(84/6/5/5) | 95(95/4/0/1) | 98 | 93(79/0/14/7) |
| 21 | 64 | 50 | 59 | 65 | 34 |
| 12 | 33 | 37 | 21 | 23 | 25 |
| 50(0/0/50/50) | 44(0/0/44/56) | 42(0/0/42/58) | 49(14/20/35/31) | 50 | 50(1/0/49/50) |
| 65(62/35/3/0) | 88(0/0/88/12) | 76(38/19/38/5) | 80(77/20/3/0) | 86 | 81(32/11/49/8) |
| 78(37/10/41/12) | 96(0/0/96/4) | 94(0/0/94/6) | 74(53/20/21/6) | 77 | 88(0/0/88/12) |
| 82(25/7/57/11) | 87(0/0/87/13) | 82(0/0/82/18) | 87(50/8/37/5) | 94 | 82(24/5/58/13) |
| 91(40/5/51/4) | 97(0/0/97/3) | 98(15/0/83/2) | 97(88/2/9/1) | 96 | 99(20/0/79/1) |
| 81(79/19/2/0) | 78(0/0/78/22) | 79(67/17/12/4) | 78(78/19/0/3) | 81 | 79(27/12/52/9) |
| 73(37/12/36/15) | 74(0/0/74/26) | 73(19/5/54/22) | 70(59/27/11/3) | 77 | 72(3/2/69/26) |
| 91(82/8/9/1) | 89(0/0/89/11) | 89(49/6/40/5) | 89(86/11/3/0) | 86 | 90(32/4/58/6) |
| 0.845 | 0.794 | 0.835 | 0.857 | 0.882 | 0.869 |

## L.2  GPT-4O-MINI

Table 8: Experimental results of GPT-4o-mini. Each item comprises the ratios of total success rate (code correct, code wrong, text correct, text wrong).

| TASK (SUCCESS RATE %) | BASELINE METHODS | | | |
|---|---|---|---|---|
| | 1.ONLY QUESTION | 2.ALL TEXT | 3.ALL CODE | 4.ALL CODE + COT |
| NUMBER MULTI. | 15(0/0/15/85) | 26(0/0/26/74) | 100(100/0/0/0) | 100(100/0/0/0) |
| GAME 24 | 15(0/0/15/85) | 16(0/0/16/84) | 9(9/91/0/0) | 10(10/84/0/6) |
| PATH PLAN | 55(49/29/6/16) | 21(0/0/21/79) | 58(58/42/0/0) | 49(49/51/0/0) |
| LETTERS | 7(0/0/7/93) | 78(0/0/78/22) | 100(100/0/0/0) | 100(100/0/0/0) |
| BOXLIFT | 37.6 | 41.7 | 40.5 | 26.4 |
| BOXNET | 10.76 | 21.94 | 20.21 | 0 |
| BLOCKSWORLD | 17(0/0/17/83) | 38(0/0/38/62) | 17(17/83/0/0) | 40(40/60/0/0) |
| DATE UNDE. | 80(0/0/80/20) | 85(0/0/85/15) | 57(57/43/0/0) | 70(70/30/0/0) |
| WEB OF LIES | 98(0/0/98/2) | 81(0/0/81/19) | 70(70/30/0/0) | 93(93/7/0/0) |
| LOGICAL DEDU. | 78(0/0/78/22) | 80(0/0/80/20) | 67(67/33/0/0) | 73(73/27/0/0) |
| NAVIGATE | 96(0/0/96/4) | 90(0/0/90/10) | 89(87/9/2/2) | 85(85/15/0/0) |
| GSM-HARD | 73(0/0/73/27) | 72(0/0/72/28) | 77(77/23/0/0) | 80(80/20/0/0) |
| MATH GEO. | 73(0/0/73/27) | 72(0/0/72/28) | 72(72/28/0/0) | 74(74/26/0/0) |
| MATH COUNT. & PROB. | 88(0/0/88/12) | 92(0/0/92/8) | 78(78/22/0/0) | 83(83/17/0/0) |
| AVERAGE NORM. SCORE | 0.676 | 0.759 | 0.774 | 0.766 |

| BASELINE METHODS | | | PROPOSED METHODS | | |
|---|---|---|---|---|---|
| 5.AUTOGEN CONCA. | 6.AUTOGEN SYSTEM | 7.CODE INTERPRETER | 8.CODE INTERPRETER+ | 9.CODE+T EXT+SUM. | 10.SELF-ESTI MATE SCORE |
| 1(0/98/1/1) | 15(0/0/15/85) | 100(100/0/0/0) | 100(100/0/0/0) | 99 | 42(31/2/11/56) |
| 13(4/24/9/63) | 14(1/4/13/82) | 62(62/38/0/0) | 83(83/17/0/0) | 17 | 23(0/3/23/74) |
| 51(51/49/0/0) | 57(54/28/3/15) | 26(26/74/0/0) | 26(26/74/0/0) | 37 | 37(34/57/3/6) |
| 100(100/0/0/0) | 7(0/0/7/93) | 87(87/13/0/0) | 89(89/11/0/0) | 90 | 51(0/0/51/49) |
| 37.4 | 39.4 | 44.7 | 64.8 | 42.8 | 38.2 |
| 16.87 | 13.18 | 23.78 | 4.17 | 22.36 | 23.34 |
| 40(0/0/40/60) | 15(0/0/15/85) | 17(3/20/14/63) | 23(7/56/16/21) | 38 | 34(0/0/34/66) |
| 63(49/35/14/2) | 80(0/0/80/20) | 74(74/26/0/0) | 77(73/23/4/0) | 83 | 82(0/0/82/18) |
| 76(0/0/76/24) | 96(0/0/96/4) | 59(42/38/17/3) | 52(30/33/22/15) | 82 | 83(0/0/83/17) |
| 75(2/0/73/25) | 76(0/0/76/24) | 75(75/25/0/0) | 78(56/17/22/5) | 82 | 73(0/0/73/27) |
| 55(0/0/55/45) | 95(0/0/95/5) | 94(94/6/0/0) | 96(84/4/12/0) | 95 | 94(0/0/94/6) |
| 68(65/32/3/0) | 73(0/0/73/27) | 73(0/0/73/27) | 52(52/48/0/0) | 77 | 73(0/0/73/27) |
| 74(29/6/45/20) | 76(0/0/76/24) | 77(0/0/77/23) | 81(74/17/7/2) | 72 | 74(0/0/74/26) |
| 88(66/9/22/3) | 88(0/0/88/12) | 83(0/0/83/17) | 87(85/12/2/1) | 88 | 87(1/0/86/13) |
| 0.718 | 0.682 | 0.808 | 0.790 | 0.850 | 0.765 |

## L.3 GPT-3.5

Table 9: Experimental results of GPT-3.5-turbo. Each item comprises the ratios of total success rate (code correct, code wrong, text correct, text wrong).

| TASK (SUCCESS RATE %) | BASELINE METHODS | | | |
|---|---|---|---|---|
| | 1.ONLY QUESTION | 2.ALL TEXT | 3.ALL CODE | 4.ALL CODE + COT |
| NUMBER MULTI. | 2(0/0/2/98) | 15(0/0/15/85) | 100(100/0/0/0) | 100(100/0/0/0) |
| GAME 24 | 3(0/0/3/97) | 4(0/0/4/96) | 12(12/88/0/0) | 9(9/91/0/0) |
| PATH PLAN | 5(1/5/4/90) | 8(0/0/8/92) | 36(36/64/0/0) | 16(16/75/0/9) |
| LETTERS | 4(0/0/4/96) | 44(0/0/44/56) | 100(100/0/0/0) | 100(100/0/0/0) |
| BOXLIFT | 37.8 | 25.2 | 21.1 | 5.3 |
| BOXNET | 6.01 | 7.35 | 0 | 0 |
| BLOCKSWORLD | 6(0/0/6/94) | 13(0/0/13/87) | 0(0/100/0/0) | 3(1/67/2/30) |
| DATE UNDE. | 61(0/0/61/39) | 53(0/0/53/47) | 50(50/50/0/0) | 38(38/62/0/0) |
| WEB OF LIES | 53(0/0/53/47) | 67(0/0/67/33) | 68(68/32/0/0) | 63(56/28/7/9) |
| LOGICAL DEDU. | 32(0/0/32/68) | 38(0/0/38/62) | 38(38/62/0/0) | 27(27/72/0/1) |
| NAVIGATE | 52(0/0/52/48) | 78(0/0/78/22) | 75(75/25/0/0) | 90(89/9/1/1) |
| GSM-HARD | 60(0/0/60/40) | 56(0/0/56/44) | 67(67/33/0/0) | 63(61/33/2/4) |
| MATH GEO. | 57(0/0/57/43) | 48(0/0/48/52) | 52(45/46/7/2) | 57(41/38/16/5) |
| MATH COUNT. & PROB. | 59(0/0/59/41) | 58(0/0/58/42) | 71(65/27/6/2) | 77(60/18/17/5) |
| **AVERAGE NORM. SCORE** | **0.653** | **0.653** | **0.6813** | **0.640** |

| BASELINE METHODS | | | PROPOSED METHODS | | |
|---|---|---|---|---|---|
| 5.AUTOGEN CONCA. | 6.AUTOGEN SYSTEM | 7.CODE INTERPRETER | 8.CODE INTERPRETER+ | 9.CODE+T EXT+SUM. | 10.SELF-ESTI MATE SCORE |
| 50(48/0/2/50) | 2(0/0/2/98) | 100(100/0/0/0) | 100(100/0/0/0) | 26 | 24(15/0/9/76) |
| 91(91/9/0/0) | 4(0/0/4/96) | 92(92/8/0/0) | 6(6/94/0/0) | 3 | 10(1/2/9/88) |
| 1(1/9/0/90) | 3(0/2/3/95) | 0(0/64/0/36) | 0(0/76/0/24) | 23 | 12(6/6/6/82) |
| 100(100/0/0/0) | 4(0/0/4/96) | 70(70/21/0/9) | 89(89/7/0/4) | 67 | 5(0/0/5/95) |
| 12.2 | 37.4 | 3.5 | 4.7 | 28.6 | 16.3 |
| 1.39 | 6.17 | 0 | 0 | 4.93 | 12.34 |
| 9(0/0/9/91) | 3(0/0/3/97) | 12(7/63/5/25) | 7(7/93/0/0) | 3 | 8(0/0/8/92) |
| 20(20/78/0/2) | 62(0/0/62/38) | 25(20/37/5/38) | 36(35/53/1/11) | 62 | 56(0/0/56/44) |
| 53(2/2/51/45) | 55(0/0/55/45) | 42(8/5/34/53) | 52(48/38/4/10) | 56 | 40(0/0/40/60) |
| 31(14/48/17/21) | 36(0/0/36/64) | 32(2/3/30/65) | 28(26/60/2/12) | 36 | 37(0/0/37/63) |
| 60(0/0/60/40) | 54(0/0/54/46) | 67(55/22/12/11) | 81(71/11/10/8) | 78 | 64(0/0/64/36) |
| 55(51/30/4/15) | 59(0/0/59/41) | 64(62/35/2/1) | 63(63/31/0/6) | 62 | 60(1/1/59/39) |
| 66(31/8/35/26) | 56(0/0/56/44) | 59(21/26/38/15) | 53(29/35/24/12) | 43 | 44(0/0/44/56) |
| 76(46/8/30/16) | 60(0/0/60/40) | 66(38/23/28/11) | 66(32/23/34/11) | 57 | 60(4/0/56/40) |
| **0.646** | **0.555** | **0.645** | **0.585** | **0.639** | **0.592** |

## L.4 O1-PREVIEW

Table 10: Experimental results of O1-preview. Each item comprises the ratios of total success rate (code correct, code wrong, text correct, text wrong).

| TASK (SUCCESS RATE %) | BASELINE METHODS | | | |
|---|---|---|---|---|
| | 1.ONLY QUESTION | 2.ALL TEXT | 3.ALL CODE | 4.ALL CODE + CoT |
| GAME 24 | 78(0,0,78,22) | 69(0,0,69,31) | 82(58,12,24,6) | 87(67,8,20,5) |
| PATH PLAN | 56(52,37,4,7) | 61(3,3,58,36) | 59(59,41,0,0) | 64(64,36,0,0) |
| BOXLIFT | 67.09 | 56.24 | 85.88 | 91.57 |
| BOXNET | 67.34 | 59.92 | 63.68 | 49.55 |
| BLOCKSWORLD | 77(0,0,77,23) | 72(0,0,72,28) | 78(27,9,51,13) | 77(31,8,46,15) |
| AVERAGE NORM. SCORE | 0.8820 | 0.8195 | 0.9331 | 0.9283 |

| BASELINE METHODS | PROPOSED METHODS | |
|---|---|---|
| 5.AUTOGEN CONCA. | 9.CODE+TEXT+SUM. | 10.SELF-ESTIMATE SCORE |
| 69(1,7,68,24) | 77 | 63(1,3,62,34) |
| 56(25,26,31,18) | 61 | 47(24,33,23,20) |
| 73.92 | 72.08 | 38.48 |
| 48.59 | 62.81 | 64.34 |
| 85(0,0,85,15) | 81 | 79(0,1,79,20) |
| 0.8394 | 0.9022 | 0.7527 |

## L.5 Claude-3-sonnet-20240229

Table 11: Experimental results of Claude-3-sonnet-20240229. Each item comprises the ratios of total success rate (code correct, code wrong, text correct, text wrong).

| Task (success rate %) | Baseline Methods | | | |
| --- | --- | --- | --- | --- |
| | 1.Only Question | 2.All Text | 3.All Code | 4.All Code + CoT |
| Number Multi. | 37(0/0/37/63) | 32(0/0/32/68) | 100(100/0/0/0) | 99(99/1/0/0) |
| Game 24 | 4(0/0/4/96) | 4(0/0/4/96) | 4(4/96/0/0) | 34(34/66/0/0) |
| Path plan | 33(4/6/29/61) | 26(0/0/26/74) | 58(58/42/0/0) | 42(42/58/0/0) |
| Letters | 2(0/0/2/98) | 9(0/0/9/91) | 100(100/0/0/0) | 100(100/0/0/0) |
| BoxLift | 48.02 | 50.31 | 12.23 | 8.19 |
| BoxNet | 21.36 | 21.01 | 24.26 | 30.50 |
| Blocksworld | 23(0/0/23/77) | 26(0/0/26/74) | 6(0/14/6/80) | 21(3/3/19/75) |
| Date Unde. | 73(0/0/73/27) | 71(0/0/71/29) | 54(54/46/0/0) | 63(63/37/0/0) |
| Web of Lies | 88(0/0/88/12) | 89(0/0/89/11) | 86(86/14/0/0) | 59(58/40/2/0) |
| Logical Dedu. | 60(0/0/60/40) | 61(0/0/61/39) | 42(42/58/0/0) | 53(53/47/0/0) |
| Navigate | 76(0/0/76/24) | 79(0/0/79/21) | 89(89/11/0/0) | 94(94/6/0/0) |
| GSM | 70(0/0/70/30) | 72(0/0/72/28) | 76(76/24/0/0) | 77(77/23/0/0) |
| MATH Geo. | 44(0/0/44/56) | 44(0/0/44/56) | 39(39/61/0/0) | 37(37/63/0/0) |
| MATH Count.&Prob. | 65(2/0/63/35) | 63(0/0/63/37) | 76(76/24/0/0) | 77(77/23/0/0) |
| **Average Norm. Score** | **0.715** | **0.720** | **0.747** | **0.810** |

| Baseline Methods | | Proposed Methods | |
| --- | --- | --- | --- |
| 5.AutoGen Conca. | 6.AutoGen System | 9.Code+Text+Sum. | 10.Self-estimate score |
| 100(100/0/0/0) | 37(0/0/37/63) | 95 | 87(82/0/5/13) |
| 5(5/95/0/0) | 4(0/0/4/96) | 7 | 69(69/31/0/0) |
| 56(56/44/0/0) | 33(4/6/29/61) | 54 | 41(41/59/0/0) |
| 99(99/0/0/1) | 2(0/0/2/98) | 74 | 34(26/0/8/66) |
| 5.05 | 48.02 | 28.79 | 0.51 |
| 28.82 | 21.36 | 19.95 | 1.89 |
| 7(6/81/1/12) | 23(0/0/23/77) | 19 | 15(0/54/15/31) |
| 60(56/34/4/6) | 73(0/0/73/27) | 72 | 68(26/19/42/13) |
| 58(48/35/10/7) | 86(0/0/86/14) | 54 | 82(0/0/82/18) |
| 55(55/45/0/0) | 62(0/0/62/38) | 61 | 60(2/4/58/36) |
| 74(62/21/12/5) | 76(0/0/76/24) | 80 | 78(0/0/78/22) |
| 68(63/29/5/3) | 70(0/0/70/30) | 69 | 62(18/25/44/13) |
| 47(43/47/4/6) | 42(0/0/42/58) | 39(9/11/30/50) | 42(21/36/21/22) |
| 77(72/20/5/3) | 64(2/0/62/36) | 65 | 62(24/11/38/27) |
| **0.740** | **0.711** | **0.762** | **0.694** |

## L.6 OPEN-MIXTRAL-8X7B

Table 12: Experimental results of Open-mixtral-8x7b. Each item comprises the ratios of total success rate (code correct, code wrong, text correct, text wrong).

| TASK (SUCCESS RATE %) | BASELINE METHODS | | | |
|---|---|---|---|---|
| | 1. ONLY QUESTION | 2. ALL TEXT | 3. ALL CODE | 4. ALL CODE + COT |
| NUMBER MULTI. | 5(0/0/5/95) | 5(0/0/5/95) | 100(100/0/0/0) | 94(94/5/0/1) |
| GAME 24 | 4(0/0/4/96) | 2(0/0/2/98) | 1(1/99/0/0) | 2(2/94/0/4) |
| PATH PLAN | 5(0/1/5/94) | 11(0/0/11/89) | 16(16/84/0/0) | 8(8/92/0/0) |
| LETTERS | 1(0/0/1/99) | 3(0/0/3/97) | 82(82/18/0/0) | 95(95/5/0/0) |
| BOXLIFT | 28.13 | 22.69 | 2.32 | 4.83 |
| BOXNET | 5.15 | 5.99 | 3.13 | 0 |
| BLOCKSWORLD | 3(0/0/3/97) | 8(0/0/8/92) | 0(0/100/0/0) | 0(0/100/0/0) |
| DATE UNDE. | 48(0/0/48/52) | 45(0/0/45/55) | 50(50/50/0/0) | 56(56/40/0/4) |
| WEB OF LIES | 53(0/0/53/47) | 66(0/0/66/34) | 56(56/42/0/2) | 61(61/38/0/1) |
| LOGICAL DEDU. | 36(0/0/36/64) | 41(0/0/41/59) | 32(32/68/0/0) | 34(32/61/2/5) |
| NAVIGATE | 58(0/0/58/42) | 41(0/0/41/59) | 58(57/42/1/0) | 69(65/30/4/1) |
| GSM | 50(0/0/50/50) | 48(0/0/48/52) | 60(60/40/0/0) | 62(61/33/1/5) |
| MATH GEO. | 42(0/0/42/58) | 44(0/0/44/56) | 46(44/49/2/5) | 49(47/45/2/6) |
| MATH COUNT.&PROB. | 50(0/0/50/50) | 54(0/0/54/46) | 74(73/25/1/1) | 64(60/35/4/1) |
| AVERAGE NORM. SCORE | 0.6345 | 0.6809 | 0.6923 | 0.6734 |

| BASELINE METHODS | | PROPOSED METHODS | |
|---|---|---|---|
| 5. AUTOGEN CONCA. | 6. AUTOGEN SYSTEM | 9. CODE+T EXT+SUM. | 10. SELF-ESTI MATE SCORE |
| 93(93/5/0/2) | 5(0/0/5/95) | 68 | 59(51/38/8/3) |
| 2(1/80/1/18) | 4(0/0/4/96) | 6 | 2(2/99/0/0) |
| 19(19/76/0/5) | 5(0/1/5/94) | 11 | 4(2/11/2/85) |
| 86(86/13/0/1) | 1(0/0/1/99) | 84 | 4(2/1/2/95) |
| 2.15 | 28.13 | 10.50 | 10.33 |
| 5.77 | 5.15 | 1.19 | 3.54 |
| 0(0/90/0/10) | 3(0/0/3/97) | 3 | 0(0/0/0/100) |
| 47(44/45/3/8) | 49(0/0/49/51) | 47 | 36(0/1/35/64) |
| 62(41/13/21/25) | 52(0/0/52/48) | 49 | 56(0/0/56/44) |
| 34(18/35/16/31) | 37(0/0/37/63) | 44 | 12(0/0/12/88) |
| 42(28/41/14/17) | 58(0/0/58/42) | 56 | 40(0/0/40/60) |
| 53(49/43/4/4) | 50(0/0/50/50) | 56 | 45(0/0/45/55) |
| 40(31/41/9/19) | 44(0/0/44/56) | 49 | 48(0/0/48/52) |
| 65(58/29/7/6) | 52(0/0/52/48) | 68 | 51(15/16/36/23) |
| 0.7084 | 0.6412 | 0.7361 | 0.4909 |

