# OpenReview forum: "Steering Large Language Models between Code Execution and Textual Reasoning"
_ICLR.cc/2025/Conference — ICLR 2025 Poster_

### Official Review · Reviewer_F65d · 2024-10-22

**Soundness:** 3
**Presentation:** 3
**Contribution:** 2
**Rating:** 6
**Confidence:** 4

**Summary:**

This paper explores if querying LLMs to generate code can be more effective than textual reasoning for tasks involving math, logic, and optimization.
Experiments across 14 tasks with 6 different LLMs show that (1) there is no single best method to optimally decide when to use code or text reasoning, (2) forcing code reasoning all the time can deteriorate performance, (3) smaller-size models tend to use code more often and outperform larger-size models for some tasks, and (4) mixing code and textual reasoning together improves performance but is limited by the model capacity and is more expensive to run due to more predicted tokens.

**Strengths:**

This is a great case study on the usage of code to solve mathematical and logical tasks with LLMs. The experimental settings are clear and easy to follow. The paper shows a lot of experimental results with various models and tasks.
The experiments are well suited to answer the paper’s investigation.
The proposed prompting method performs better on average than all other methods tried, at the expense of generated token length.
Experimental results should be of interest to the research community.

**Weaknesses:**

This paper is a technical analysis of various prompting models & methods on math and logical reasoning tasks. While the results are interesting and the experiments exhaustive, this work does not provide a lot of solutions. The only novel contribution of this paper (aside from the experimental results) is the 3 proposed prompting methods: CodeInterpreter+, Code+Text+Sum., Self-Estimate Score, among which only one (Code+Text+Sum.) seems to perform well across the evaluated tasks. This work could benefit from a more detailed investigation on particular tasks to help identify potential solutions. For instance, analyzing the probabilistic confidence of the model when generating a text answer rather than code.

The paper claims that all tasks tested in this paper can be solved with code (with varying difficulty), yet no code-generation model has been tried. I strongly suggest the authors to add an experiment with a code generation model. Although the only prompting method that would make sense here is the default “Only Question”, it is an important baseline to consider.

Minor: the fact gpt3.5 outperforms gpt4 in some tasks does not mean there is an “inverse scaling _law_”. This is a very specific case in which smaller models are less certain and thus use external tools (code interpreters), which is expected.

**Questions:**

In general, the approach of trying to find the prompting method to solve all these tasks could never end. Prompt engineering can be much more effective when targeted to single tasks as one can give more information about the task, in context examples, etc.
Did you try task-specific prompts? Do you think you could solve some of these tasks like this?

After finding that large models generate code that simulates text reasoning rather than actual code, did you try to further improve the prompt? Simple things to try would be to add phrases like "think of an algorithm to solve the task and implement it in python", "do not try to answer in one step",  "be careful this is hard, be less confident" etc...

---

> ### Author Response · Authors · 2024-11-20
> **Response to the Reviewer F65d - Question 1**
>
> Thank you for the valuable feedback and insightful comments. It seems there were some misunderstandings regarding our work and its perceived weaknesses. We aim to **address these misunderstandings in detail** and **have added three experiments** based on reviewers’ suggestions. We kindly hope the reviewer will reconsider our work in light of the responses provided below. **The revised paper has been uploaded with the changed contents blue colored.**
>
> ***Question 1:*** *This work does not provide a lot of solutions. The only novel contribution of this paper (aside from the experimental results) is the 3 proposed prompting methods: CodeInterpreter+, Code+Text+Sum., Self-Estimate Score, among which only one (Code+Text+Sum.) seems to perform well across the evaluated tasks.*
>
> **Response 1:**
>
> * **We believe the contributions of this paper are enough**. Apart from the proposed methods, we emphasize the significance of steering LLMs for text/code generation, test the limitations of existing methods, uncover phenomena like inverse scaling behavior, evolution with complexity, and different types of code answers, and reveal the underlying mechanism, as stated in the original paper line 97-116.
>
> * **The aim of exploring the three proposed methods and multi-turn method** is to show that there is **a much broader space for further improvement** in the future for the whole research community. Even the simple techniques like Code+Text+Sum. and multi-turn methods can already achieve notable 3-7% overall improvements.
>
> * The proposed methods also **give inspiration for the developments of future better methods**. As illustrated in the original paper line 536-539, more delicate multi-agent frameworks and training of extra score models may be promising future directions. Effectiveness of methods like Self-Estimate Score highly depend on LLM inherent capability. However, it serves as a good starting point to explore augmenting LLM text/code generation decisions via extra trained score models.
>
> * **Developing a final solution is very challenging.** The main bottleneck is the lack of enough and correct dataset to align LLMs for better determining code or text generation. Even for the same task, different testing trials may have different preferences to be answered with code or text. Meanwhile, how to develop a generalizable method is hard. Training on limited domains will not be applicable to others. Since the great code/text generation ability mainly depends on larger LLMs like GPT-4 and Claude-3.5, directly training on these close models is also unavailable. Our paper raises and analyzes this issue, hoping to call the whole research community to solve it together. We will continue to explore the final solution.

---

> > ### Author Response · Authors · 2024-11-20
> > **Response to the Reviewer F65d - Question 2-4**
> >
> > ***Question 2:*** *A more detailed investigation on particular tasks to help identify potential solutions. For instance, analyzing the probabilistic confidence of the model when generating a text answer rather than code.*
> >
> > **Response 2:** Thanks for the insightful advice, **here we added the experiments to compare and analyze the probabilistic confidence of the model when generating text or code answers**. We tested on GPT-4o, GPT-4o-mini, and GPT-3.5, collecting the log p values of each output token supported by OpenAI API and calculating the perplexity (Perplexity(W) = P(w₁w₂...wₙ)^(-1/n)) of each generated response. Since we can only access the token probability of the LLMs without Code Interpreter, the Only Question setting will always generate text responses without extra prompt guidance. To stimulate the LLMs to generate answers with either text or code modes, we compare three settings with varied prompts: Only Question, All Code, All Code + CoT.
> >
> > As shown in the following table and Figure 13 in the revised paper, the perplexity in three settings are close. The All Code + CoT setting has slightly higher perplexity compared to other two settings in GPT-4o and GPT-4o-mini. However, this difference in perplexity has no notable impact on success rates. This phenomenon is reasonable since LLMs are trained to use perplexity as the metric and try to decrease it, meaning the code/text difference will not be shown in direct perplexity comparison if the original training process does not deliberately distinguish text and code. We have added these experiments and the related discussion in the revised paper (Line 525-527, 1404-1445).
> >
> > | Model | Task | Only Question (Perplexity/Success) | All Code + CoT (Perplexity/Success) | All Code (Perplexity/Success) |
> > |-------|------|------------------------|------------------------|------------------------|
> > | GPT-4o | Number Multiply | 1.11 / 0.37 | 1.18 / 1.00 | 1.01 / 1.00 |
> > | GPT-4o | Game24 | 1.13 / 0.17 | 1.34 / 0.05 | 1.21 / 0.11 |
> > | GPT-4o | BoxLift | 1.18 / 0.69 | 1.21 / 0.30 | 1.14 / 0.68 |
> > | GPT-4o-mini | Number Multiply | 1.09 / 0.15 | 1.14 / 1.00 | 1.01 / 1.00 |
> > | GPT-4o-mini | Game24 | 1.12 / 0.15 | 1.33 / 0.09 | 1.14 / 0.10 |
> > | GPT-4o-mini | BoxLift | 1.09 / 0.38 | 1.22 / 0.40 | 1.11 / 0.26 |
> > | GPT-3.5 | Number Multiply | 1.43 / 0.02 | 1.10 / 1.00 | 1.00 / 1.00 |
> > | GPT-3.5 | Game24 | 1.33 / 0.03 | 1.11 / 0.12 | 1.20 / 0.09 |
> > | GPT-3.5 | BoxLift | 1.10 / 0.38 | 1.10 / 0.21 | 1.06/ 0.05 |
> >
> > ***Question 3:*** *Strongly suggest the authors to add an experiment with a code generation model.*
> >
> > **Response 3:** Thank you for the helpful suggestions, **we have added the experiments to test on specialized Code LLMs: CodeLlama and Qwen2.5-Coder-32B-Instruct (the current best Code LLM on leaderboard)**. As shown in the following table, even for specialized code LLM, the problem of determining whether to generate code or text still influences the LLM performance quite a lot. In Only Question setting, both CodeLlama and Qwen2.5-Coder-32B-Instruct will mostly generate direct text answers without extra prompt guidance, which achieves much lower success rates compared to code answers in other two settings in Number Multiply task. Meanwhile, the overall performance of CodeLlama-34b-Instruct-hf and Qwen2.5-Coder-32B-Instruct are not as good as GPT-4o, showing the code LLMs may overfit to trained code datasets. We have added these experimental results and related discussion in the revised paper (Line 523-525, 1356-1377).
> >
> > | Model | Task | Only Question (Success) | All Code (Success) | All Code + CoT (Success) |
> > |-------|------|----------------|-----------------|-----------------|
> > | CodeLlama-34B | Number Multiply | 0.00 | 0.49 | 0.55 |
> > | CodeLlama-34B | Game24 | 0.01 | 0.01 | 0.00 |
> > | CodeLlama-34B | BoxLift | 0.37 | 0.25 | 0.20 |
> > | Qwen2.5-Coder-32B-Instruct | Number Multiply | 0.21 | 0.99 | 1.00 |
> > | Qwen2.5-Coder-32B-Instruct | Game24 | 0.23 | 0.21 | 0.07 |
> > | Qwen2.5-Coder-32B-Instruct | BoxLift | 0.43 | 0.32 | 0.56 |
> >
> > ***Question 4:*** *The fact gpt3.5 outperforms gpt4 in some tasks does not mean there is an “inverse scaling law”. This is a very specific case in which smaller models are less certain and thus use external tools (code interpreters), which is expected.*
> >
> > **Response 4:** Thank you for the helpful advice. In the revised paper, we have used ‘inverse scaling phenomenon’ instead of ‘inverse scaling law’. We also have updated the paper to further emphasize that the phenomenon is observed for only specific tasks. Thank you!
> >
> > **The phenomenon that smaller models using Code Interpreters more is not expected**. To our best knowledge, **no previous papers and works have revealed this phenomenon**. Meanwhile, intuitively, people always expect larger models should be more capable of using tools/generating codes. This inverse scaling phenomenon is shown to exist only after real testing and seems to be reasonable after detailed analysis.

---

> > > ### Author Response · Authors · 2024-11-20
> > > **Response to the Reviewer F65d - Question 5 and 6**
> > >
> > > ***Question 5:*** *In general, the approach of trying to find the prompting method to solve all these tasks could never end. Prompt engineering can be much more effective when targeted to single tasks as one can give more information about the task, in context examples, etc. Did you try task-specific prompts? Do you think you could solve some of these tasks like this?*
> > >
> > > **Response 5:**
> > >
> > > * **We believe developing effective methods or prompts for multiple tasks should be the future**. Recent trends on the popularity of apps such as ChatGPT and Microsoft Copilot suggest a single LLM powered chat system being used for a wide variety of tasks. As such, studying all the tasks under one prompting strategy is useful and important for improving such apps.
> > >
> > > * **The effectiveness of task-specific prompts depends on the informativeness of user input hints and task characteristics**. In tasks like Game24, Number Multiply, BoxLift, and Letters, one type of python code can solve all the tested samples. In this case, if the prompt comprises the correct code as few-shot examples. The LLMs can well solve this task. However, this type of prompting is regarded as cheating since users input correct answers to the prompt and lack generalizability to multiple tasks. Meanwhile, some tasks like Date Understanding and Math do not have unified code to solve all samples. In this case, either using correct code or text as few-shot examples are the same.
> > >
> > > * **The prompts with text as few-shot examples have already been included in the current tested question prompt**. In some tasks like Game24, Math, and Path Plan, the original question prompt from the original dataset already comprises few-shot question/text answers pairs as examples. As shown in the testing experiments, the LLMs still cannot completely solve the tasks.
> > >
> > > * **The prompts without few-shot examples but finely optimized by automatic prompt optimization frameworks have been tested in other papers**. In recent works from other researchers such as PromptAgent, PromptBreeder, PROMST, they developed optimized prompts to guide LLM reasoning and tested on several tasks the same as this work. They find the optimized prompts can improve the success rate apparently (e.g., improve BoxNet from 0.65 to 0.79 for GPT-4o), but not completely solve the whole task.
> > >
> > > ***Question 6:*** *After finding that large models generate code that simulates text reasoning rather than actual code, did you try to further improve the prompt? Simple things to try would be to add phrases like "think of an algorithm to solve the task and implement it in python", "do not try to answer in one step", "be careful this is hard, be less confident" etc...*
> > >
> > > **Response 6:** **Thanks for the suggestions from the reviewer 697L and F65d, we have added further experiments to test on more prompt versions for steering LLM code generation with Code Interpreter.** The tested prompts are as follows: 1) ‘Generate pure Python code with rich natural language-annotated comments.’ 2) ‘Think of an algorithm to solve the task and implement it in python.’ 3) ‘Be careful this is hard, be less confident.’ 4) ‘Think of an algorithm to solve the task and implement it in python. Be careful this is hard, be less confident.’. We compare these prompt versions with the Original Prompt ‘Think the task step by step if you need to. If a plan is not provided, explain your plan first. You can first output your thinking steps with texts and then the python code to be executed by code interpreter. Try to use code interpreter more.’.
> > >
> > > The experimental results are shown in the following table. Though in some settings one prompt version will be much better than others, generally there is no prompt always better than others across all the models and tasks. Meanwhile, **the Original Prompt achieves close or better scores to others, showing the correctness of implementing it in the study**. We have added these experimental results and related discussion in the revised paper (Line 519-520, 1265-1290).
> > >
> > > | Prompt Version | Model | Game24 | Path Plan | BoxLift | Normalized Score |
> > > |---------------|--------|---------|------------|----------|------------------|
> > > | Original Prompt | GPT-4o | 0.63 | 0.46 | 0.59 | 0.85 |
> > > | | GPT-4o-mini | 0.83 | 0.26 | 0.65 | 0.80 |
> > > | Prompt 1 | GPT-4o | 0.63 | 0.40 | 0.47 | 0.75 |
> > > | | GPT-4o-mini | 1.00 | 0.46 | 0.34 | 0.84 |
> > > | Prompt 2 | GPT-4o | 0.91 | 0.43 | 0.67 | 0.97 |
> > > | | GPT-4o-mini | 0.74 | 0.20 | 0.31 | 0.55 |
> > > | Prompt 3 | GPT-4o | 0.64 | 0.40 | 0.62 | 0.83 |
> > > | | GPT-4o-mini | 0.77 | 0.16 | 0.46 | 0.61 |
> > > | Prompt 4 | GPT-4o | 0.88 | 0.41 | 0.68 | 0.95 |
> > > | | GPT-4o-mini | 0.94 | 0.24 | 0.51 | 0.75 |
> > >
> > > We appreciate the opportunity to engage further with the reviewer to clarify our contributions. Thank you once again for your time and thoughtful review of our work.

---

> > > > ### Comment · Reviewer_F65d · 2024-11-21
> > > > **acknowledgement**
> > > >
> > > > Thank you for the detailed answers and the updated manuscript. Below are some comments based on your responses.
> > > >
> > > > - I agree with reviewer 697L that the paper reads more like a technical report. While I recognize this work shows the importance of balancing code/text generation to solve complex tasks, since there is no generic solution proposed (which is expected when working on prompt engineering because it all depends on the task we want to achieve), the conclusion feels like: based on what one tries to achieve, it may or may not be beneficial to generate code first. Nevertheless the paper can still inspire other work to explore similar prompting methods, that is why I gave it a positive score (>5).
> > > >
> > > > - I find it surprising that code models do not generate code by default... This is probably due to the fact that their prompt is confusing them (the problem statement being written in English in their prompt maybe). It would be nice to know if we represent the problem in a pure coded template if code llms can generate code that solves or partially solve some of the tasks explored in this work. Although representing such problems in a code format may be not trivial...
> > > >
> > > > --- _The following is purely subjective and does not influence the quality of this work, it is more a discussion :)_ ---
> > > >
> > > > > We believe developing effective methods or prompts for multiple tasks should be the future. Recent trends on the popularity of apps such as ChatGPT and Microsoft Copilot suggest a single LLM powered chat system being used for a wide variety of tasks.
> > > >
> > > > I agree that developing effective **methods** that generalize across tasks should be the future. I am however doubtful that one **prompt** can achieve widely different tasks. For instance, the fact one prompt works with a language model but does not with a code model shows this. The prompt design is inherently conditioned on the LLM at hand. A prompt to achieve a specific task may be different if one uses llama or claude or gpt or mistral.
> > > >
> > > > Systems like ChatGPT are designed to be good instruction-following agents. Instruction-following is a specific task. It can be seen as a sort of meta-task, which asks the user to give a prompt to do a specific task, essentially deferring the prompt design to the user. Furthermore, these systems are continuously trained on millions of human data collected daily, they are not _just_ prompted LLMs.
> > > > It may even be the case that in the future, these systems have no use for a prompt as a lot of engineering pipelines around the LLM ensure it is "safe" and training on huge amounts of quality data makes them good "instruction followers" by default. This is all speculation but interesting to think about nonetheless.

---

> > > > > ### Author Response · Authors · 2024-11-23
> > > > > **Response to Reviewer F65d - Second Round - 1**
> > > > >
> > > > > ***Comment 1:*** *I agree with reviewer 697L that the paper reads more like a technical report. While I recognize this work shows the importance of balancing code/text generation to solve complex tasks, since there is no generic solution proposed (which is expected when working on prompt engineering because it all depends on the task we want to achieve), the conclusion feels like: based on what one tries to achieve, it may or may not be beneficial to generate code first. Nevertheless the paper can still inspire other work to explore similar prompting methods, that is why I gave it a positive score (>5).*
> > > > >
> > > > > **Response 1:** Thank you for your positive feedback. Apart from all the prompting methods, our work **aims to emphasize that solving tasks with code or text can lead to quite different performance even for the same LLM**. **Current existing methods can not decide upon code/text decisions quite well**. In our paper, we propose using prompting and agent framework to improve general performance, which truly result in notable improvements. However, to build up a generic solution, training-free methods are not enough. **We believe future work should combine model finetuning, prompt tuning, and agent framework together for a much satisfying solution. We have added this discussion in the paper Line 124-126**.
> > > > >
> > > > > ***Comment 2:*** *It would be nice to know if we represent the problem in a pure coded template if code llms can generate code that solves or partially solve some of the tasks explored in this work. Although representing such problems in a code format may be not trivial…*
> > > > >
> > > > > **Response 2:** Thank you for the nice suggestion. We have **added the experiment to convert the text templated prompts into code templated prompts. As shown in the ‘Code Prompt’ column of the following table, it works!**
> > > > >
> > > > > We directly query GPT-4 to translate the original text prompts into the code prompt that the Code LLM can understand with the prompt 'Represent the above problem description in a pure coded template to test if Code LLMs can generate code to solve this problem. Output the prompt to steer Code LLMs to answer the whole question by outputting the complete python code. Do not output direct code answers.'
> > > > >
> > > > > The example code prompt is like:
> > > > >
> > > > > ‘
> > > > > Given four numbers, use each number exactly once along with basic arithmetic operations (+, -, *, /) to form an expression that evaluates to 24. Each number must be used, and you can use parentheses to define the order of operations. Your task is to write a Python function that takes a list of four numbers as input and returns a string representing the expression that evaluates to 24. If no such expression exists, return an empty string.
> > > > > ```python
> > > > > def find_expression_to_24(numbers):
> > > > >     # Your code here to find the expression
> > > > >     return expression
> > > > > # Test cases
> > > > > print(find_expression_to_24([9, 10, 11, 13]))  # Output: "((10-9)*(11+13))"
> > > > > print(find_expression_to_24([4, 10, 10, 11]))  # Output: "((4*11)-(10+10))"
> > > > > print(find_expression_to_24([5, 6, 13, 13]))  # Output: "((5-(13/13))*6)"
> > > > > print(find_expression_to_24([2, 6, 6, 7]))    # Output: "((6+(6*7))/2)"
> > > > > print(find_expression_to_24([2, 6, 10, 18]))  # Output: "(2-(6-(10+18)))"
> > > > > print(find_expression_to_24([1, 1, 4, 6]))    # Output: "<<<answer>>>"
> > > > > ```
> > > > > Your task is to implement the `find_expression_to_24` function to solve the problem.
> > > > > ’
> > > > >
> > > > > We find that 1) With code prompt, the Code LLM generates code answers in nearly all cases. 2) In some situations, the Code LLM truly performs much better than other settings. For example, Qwen2.5-Coder-32B-Instruct performs much better with direct code prompt in Game24 task. 3) Whether code prompt works depends on the inherent capabilities of Code LLM and the task difficulty. CodeLlama-34b-Instruct-hf still does not perform notably better with code prompt. The written codes are mostly wrong. Qwen2.5-Coder-32B-Instruct still does not perform apparently better in BoxLift task, since the the required code in BoxLift is more challenging.
> > > > >
> > > > > | Model | Task | Only Question (Success) | **Code Prompt**  | All Code (Success) | All Code + CoT (Success) |
> > > > > |-------|------|----------------|----------------|-----------------|-----------------|
> > > > > | CodeLlama-34B | Number Multiply | 0.00 | **0.62** | 0.49 | 0.55 |
> > > > > | CodeLlama-34B | Game24 | **0.01** | 0.00 | **0.01** | 0.00 |
> > > > > | CodeLlama-34B | BoxLift | **0.37** | 0.22 | 0.25 | 0.20 |
> > > > > | Qwen2.5-Coder-32B-Instruct | Number Multiply | 0.21 | **1.00** | 0.99 | **1.00** |
> > > > > | Qwen2.5-Coder-32B-Instruct | Game24 | 0.23 | **0.74** | 0.21 | 0.07 |
> > > > > | Qwen2.5-Coder-32B-Instruct | BoxLift | 0.43 | 0.54 | 0.32 | **0.56** |
> > > > >
> > > > > Though the code prompt introduces hints to guide LLM to generate code all the time (different from the setting in our study), we think the above experiments do serve as a good baseline and strengthen the understandings. We have **added these new contents and discussions in the modified paper Line 1365-1416.**

---

> > > > > > ### Author Response · Authors · 2024-11-23
> > > > > > **Response to Reviewer F65d - Second Round - 2**
> > > > > >
> > > > > > ***Comment 3:*** *--- The following is purely subjective and does not influence the quality of this work, it is more a discussion :) ---
> > > > > > I agree that developing effective methods that generalize across tasks should be the future. I am however doubtful that one prompt can achieve widely different tasks. For instance, the fact one prompt works with a language model but does not with a code model shows this. The prompt design is inherently conditioned on the LLM at hand. A prompt to achieve a specific task may be different if one uses llama or claude or gpt or mistral.
> > > > > > Systems like ChatGPT are designed to be good instruction-following agents. Instruction-following is a specific task. It can be seen as a sort of meta-task, which asks the user to give a prompt to do a specific task, essentially deferring the prompt design to the user. Furthermore, these systems are continuously trained on millions of human data collected daily, they are not just prompted LLMs. It may even be the case that in the future, these systems have no use for a prompt as a lot of engineering pipelines around the LLM ensure it is "safe" and training on huge amounts of quality data makes them good "instruction followers" by default. This is all speculation but interesting to think about nonetheless.*
> > > > > >
> > > > > > **Response 3:** Thank you for your kind and insightful discussion. We totally agree with your opinions above. Pursuing a method or framework to generalize across tasks should be the future. However, we believe purely tuning prompts is not enough to realize this. Our study in tuning prompts for better text/code decision and many other studies have all shown that current LLMs have preference to prompts. No prompt is generalizable across tasks and LLMs. That is why in our work we also implemented agent frameworks to see whether it can perform better. It truly performs better, while we also believe that is not the final solution.
> > > > > >
> > > > > > The final solution should combine model training, agent framework, and prompt tuning together. However, there are many challenges such as lack of reasonable training frameworks and diverse datasets. In the problem of text/code decisions, we are worried that directly tuning LLMs to answer with code or text on some tasks is not generalizable to other tasks. Another solution is to explore whether using multi-agents/LLMs framework can be better, i.e., training a separate model to generate prompts to guide the generation of task LLM. Then how to define the reward for the model on this text/code decision task is challenging. We will continue working on this.
> > > > > >
> > > > > > We are very happy to discuss more with the reviewer about this work and research insights.

---

> > > > > > ### Comment · Reviewer_F65d · 2024-11-27
> > > > > > **about code prompts**
> > > > > >
> > > > > > Thank you for running this experiment, I am very happy to see it working! :) I think it is a very important baseline that should be part of the main text. It also shows that 100% success is doable with code as you mention in the main text.

---

### Official Review · Reviewer_1VAn · 2024-10-30

**Soundness:** 3
**Presentation:** 4
**Contribution:** 3
**Rating:** 6
**Confidence:** 2

**Summary:**

The article titled "LLM CodeSteer: Steering Large Language Models Between Code Execution and Textual Reasoning" explores the balance between code execution and textual reasoning capabilities in Large Language Models (LLMs). It argues that while textual reasoning has limitations, especially in tasks involving math, logic, and optimization, direct coding can often provide a more effective solution. The study conducts experiments with 14 diverse tasks and 6 types of LLMs, finding that there is no single optimal method for guiding LLMs to choose between code generation and textual reasoning. It also observes an inverse scaling law, where smaller models sometimes outperform larger ones when augmented with a Code Interpreter (CI). The paper proposes three methods to improve LLM decisions on code/text generation: Code Interpreter+, Code + Text + Sum., and Self-estimate Score. These methods aim to enhance performance by combining code and textual reasoning or by using multi-turn execution/refinement. The article concludes that guiding LLMs in code/text generation is crucial for developing more capable agents and that there is significant room for improvement in current methods.

**Strengths:**

1. Comprehensive Analysis: The study provides a thorough analysis of LLMs' performance across a wide range of tasks, offering valuable insights into their strengths and weaknesses in code execution versus textual reasoning.

2. Sound Method: Practical Approach: By focusing on real-world tasks that can be solved through coding, the research offers practical applications for enhancing LLMs' capabilities. The proposed methods, such as Code + Text + Sum. and Self-estimate Score, present innovative ways to improve LLMs' decision-making processes in choosing between code and text.

3. This paper attempts to address an important question and proposes an effective method that achieves better performance than the compared methods.

4. The paper is well-organized and clearly written.

**Weaknesses:**

1. Dependence on Task Type: The effectiveness of code versus textual reasoning is highly dependent on the task at hand, which may limit the generalizability of the findings.

2. Overconfidence Issue: The study highlights that larger models tend to be overconfident in their textual reasoning abilities, which can lead to suboptimal performance when code execution is more effective.

**Questions:**

None

---

> ### Author Response · Authors · 2024-11-20
> **Response to the Reviewer 1VAn**
>
> Thank you for your appreciation with our work and helpful suggestions. The following are our responses to your questions, the added experiments, and the related modifications of the paper. Hope the reviewer could kindly re-evaluate our work based on our modifications and new experimental results. **The revised paper has been uploaded with the changed contents blue colored.**
>
> ***Question 1:*** *Dependence on Task Type: The effectiveness of code versus textual reasoning is highly dependent on the task at hand, which may limit the generalizability of the findings.*
>
> **Response 1:** **Our finding that either coding or textual reasoning is not always better across tasks and models is the contribution of our study rather than a limitation.** One of our contributions is the finding that currently there is no optimal method across the tasks to guide LLMs to generate code to answer the question when needed. The above findings are general to all current LLMs and tasks.
>
> ***Question 2:*** *Overconfidence Issue: The study highlights that larger models tend to be overconfident in their textual reasoning abilities, which can lead to suboptimal performance when code execution is more effective.*
>
> **Response 2:** **This inverse scaling behavior is an interesting phenomenon found in our study, which should be regarded as our contribution but not weakness.** This phenomenon reveals the limitation of current LLMs and methods to steer code/text generation. Though currently we hypothesize that this phenomenon is due to the overconfidence of larger LLMs, we believe more research can be done to understand more about this bottleneck and phenomenon in future study.
>
> **Apart from the above clarifications, we also have added more experiments on 1) testing the difference of log probability for output tokens for text or code responses. 2) testing on specialized code LLMs as baselines. 3) doing ablation studies on more prompt templates for steering LLM code generation.** These experimental results have been added to the revised paper (Line 519-527, 1265-1290, 1356-1445).
>
> We look forward to further communications with the reviewer to clarify our contributions. Thanks again for your time and patience for reviewing our work.

---

> > ### Comment · Reviewer_1VAn · 2024-11-25
> >
> > Thanks for your reply, I want to keep my overall rating.

---

### Official Review · Reviewer_697L · 2024-11-03

**Soundness:** 4
**Presentation:** 4
**Contribution:** 3
**Rating:** 6
**Confidence:** 4

**Summary:**

This paper benchmarks seven prompting methods that combine text and code to tackle 14 tasks using six strong LLMs. The results indicate no universally optimal prompting method across all tasks and models, leading the authors to propose three new prompting methods that yield consistent improvements.

**Strengths:**

- The paper effectively identifies and analyzes the challenges faced by state-of-the-art LLMs in determining when to leverage code and uncovers some interesting phenomena. These findings offer valuable guidance for balancing code and text in LLM prompting.
- The proposed prompting methods are simple yet demonstrate effectiveness across different models.
- The paper is well-structured and easy to follow, with charts and tables that greatly enhance readability.

**Weaknesses:**

The paper reads more as a technical report than a top-tier scientific contribution.
- From a scientific perspective, the problem, while useful as a testbed for prompt engineering with code and text, is not significant enough. If framed as a technical report, a broader workload and richer insights might be expected to better inform the community. While the prompting comparisons are comprehensive, they are relatively straightforward, meaning the work may not substantially save time for others. Insights largely remain at the level of observed phenomena, describing model behavior (e.g., line 274: "prompting LLMs to always respond with code (Fig 6b) performs just as poorly, or even worse"). Additional investigation into the reasons behind these behaviors might enhance understanding.
- Additionally, the three proposed methods appear to be ensembles of existing approaches, such as summarization and self-reflection. Though effective, these methods do not introduce significant new insights.

**Questions:**

- I am curious about how to delineate the boundary between text and code in prompting. For instance, if the model is prompted to generate "pure Python code with rich natural language-annotated comments," which category does this fall under? If it’s considered "All code," could we perhaps frame all tasks this way, where the simplest output might be `print(answer)`?

---

> ### Author Response · Authors · 2024-11-20
> **Response to the Reviewer 697L - Question 1**
>
> Thank you for the helpful reviews and comments. It appears there may have been a miscommunication regarding our contribution. We have, therefore, improved the description to **clarify the misunderstandings** and **added three experiments suggested by reviewers**. We hope the reviewer could kindly re-evaluate our work based on our responses provided below. **The revised paper has been uploaded with the changed contents blue colored.**
>
> ***Question 1:*** *From a scientific perspective, the problem, while useful as a testbed for prompt engineering with code and text, is not significant enough.*
>
> **Response 1:**
>
> * **Integrating textual reasoning and symbolic computing/coding for LLM response is important.** As stated in the original paper line 15-17, line 74-82, and line 864-882 Figure 9, from a scientific perspective, **one significant contribution of our study is to reveal the importance of steering LLMs to generate code** to solve questions when needed. Textual reasoning has inherent limitations in solving tasks with challenges in math, logics, optimization, and searching, which is unlikely to be solved by simply scaling up the model and data size. Coding is a necessary way to enhance LLM capability. The significance of combining symbolic computing with LLMs has also been underlined in other recent works, such as ‘LLMs can't plan, but can help planning in LLM-modulo frameworks, ICML 2024’.
>
> * **The uncovered phenomena like inverse scaling behavior, evolution with complexity, and different types of code answers are novel.** No previous studies have revealed these phenomena while they do impact LLM performance quite a lot. The phenomena like **inverse scaling behavior and varied code types are counter-intuitive but proven to be reasonable** after detailed analysis, which will help improve the ability of future LLMs and methods.
>
> * **Another contribution is to show that currently there is no optimal method on steering LLM text/code generation**, based on our testing on 7 baseline methods across 14 tasks and 6 LLMs. **We also show that there is a much broader space for further improvement.** The three proposed methods and the multi-turn method are not supposed to be the optimal solution, but to show that even the simple techniques can already notably improve LLM performance on determining code/text generation.

---

> ### Author Response · Authors · 2024-11-20
> **Response to the Reviewer 697L - Question 2 and 3**
>
> ***Question 2:*** *If framed as a technical report, a broader workload and richer insights might be expected to better inform the community. Additional investigation into the reasons behind these behaviors might enhance understanding.*
>
> **Response 2:**
>
> * **Many detailed studies have been carried out in the original paper**, to reveal underlying mechanisms such as inverse scaling behavior with varied model sizes and confidence, evolution with task complexity, different code types, and different prompting versions.
>
> * For richer insights, **here we added the experiments to compare and analyze the probabilistic confidence of the model when generating text or code answers**. We tested on GPT-4o, GPT-4o-mini, and GPT-3.5, collecting the log p values of each output token supported by OpenAI API and calculating the perplexity (Perplexity(W) = P(w₁w₂...wₙ)^(-1/n)) of each generated response. Since we can only access the token probability of the LLMs without Code Interpreter, the Only Question setting will always generate text responses without extra prompt guidance. To stimulate the LLMs to generate answers with either text or code modes, we compare three settings with varied prompts: Only Question, All Code, All Code + CoT.
>
>   As shown in the following table and Figure 13 in the revised paper, the perplexity in three settings are close. The All Code + CoT setting has slightly higher perplexity compared to other two settings in GPT-4o and GPT-4o-mini. However, this difference in perplexity has no notable impact on success rates. This phenomenon is reasonable since LLMs are trained to use perplexity as the metric and try to decrease it, meaning the code/text difference will not be shown in direct perplexity comparison if the original training process does not deliberately distinguish text and code. We have added these experiments and the related discussion in the revised paper (Line 525-527, 1404-1445).
>
> | Model | Task | Only Question (Perplexity/Success) | All Code + CoT (Perplexity/Success) | All Code (Perplexity/Success) |
> |-------|------|------------------------|------------------------|------------------------|
> | GPT-4o | Number Multi. | 1.11 / 0.37 | 1.18 / 1.00 | 1.01 / 1.00 |
> | GPT-4o | Game24 | 1.13 / 0.17 | 1.34 / 0.05 | 1.21 / 0.11 |
> | GPT-4o | BoxLift | 1.18 / 0.69 | 1.21 / 0.30 | 1.14 / 0.68 |
> | GPT-4o-mini | Number Multi. | 1.09 / 0.15 | 1.14 / 1.00 | 1.01 / 1.00 |
> | GPT-4o-mini | Game24 | 1.12 / 0.15 | 1.33 / 0.09 | 1.14 / 0.10 |
> | GPT-4o-mini | BoxLift | 1.09 / 0.38 | 1.22 / 0.40 | 1.11 / 0.26 |
> | GPT-3.5 | Number Multi. | 1.43 / 0.02 | 1.10 / 1.00 | 1.00 / 1.00 |
> | GPT-3.5 | Game24 | 1.33 / 0.03 | 1.11 / 0.12 | 1.20 / 0.09 |
> | GPT-3.5 | BoxLift | 1.10 / 0.38 | 1.10 / 0.21 | 1.06/ 0.05 |
>
> * For further understanding the significance of reasonable text/code generation, **we also added the experiments to test on specialized Code LLMs: CodeLlama-34b-Instruct-hf and Qwen2.5-Coder-32B-Instruct (the current best Code LLM on leaderboard)**. As shown in the following table, even for specialized code LLM, the problem of determining whether to generate code or text still influences the LLM performance quite a lot. In Only Question setting, both CodeLlama and Qwen2.5-Coder-32B will mostly generate direct text answers without extra prompt guidance, which achieves much lower success rates compared to code answers in other two settings in Number Multiply task. We also convert text prompts into code prompts (Code Prompt setting) and find the Code LLMs can perform better. The overall performance of CodeLlama and Qwen2.5-Coder-32B are not as good as GPT-4o, showing the code LLMs may overfit to trained code datasets. We have added these experimental results and related discussion in the revised paper (Line 523-525, 1365-1416).
>
> | Model | Task | Only Question (Succ.) | **Code Prompt**  | All Code (Succ.) | All Code + CoT (Succ.) |
> |-------|------|----------------|----------------|-----------------|-----------------|
> | CodeLlama-34B | Number Multi. | 0.00 | **0.62** | 0.49 | 0.55 |
> | CodeLlama-34B | Game24 | **0.01** | 0.00 | **0.01** | 0.00 |
> | CodeLlama-34B | BoxLift | **0.37** | 0.22 | 0.25 | 0.20 |
> | Qwen2.5-Coder-32B | Number Multi. | 0.21 | **1.00** | 0.99 | **1.00** |
> | Qwen2.5-Coder-32B | Game24 | 0.23 | **0.74** | 0.21 | 0.07 |
> | Qwen2.5-Coder-32B | BoxLift | 0.43 | 0.54 | 0.32 | **0.56** |
>
> ***Question 3:*** *While the prompting comparisons are comprehensive, they are relatively straightforward, meaning the work may not substantially save time for others.*
>
> **Response 3:** Several of our findings are novel and uncover new patterns, such as inverse scaling behavior, evolution with complexity, and different types of code answers. These insights can be valuable for people working on problems requiring code generation and reasoning. We would appreciate it if the reviewer could please share any specific prior work that has already covered these findings, or why they think our experiments are 'straightforward'? ".

---

> > ### Author Response · Authors · 2024-11-20
> > **Response to the Reviewer 697L - Question 4**
> >
> > ***Question 4:*** *The three proposed methods appear to be ensembles of existing approaches, such as summarization and self-reflection. Though effective, these methods do not introduce significant new insights.*
> >
> > **Response 4:**
> >
> > * **We believe the contributions of this paper are enough.** Apart from the proposed methods, we emphasize the significance of steering LLMs for text/code generation, test the limitations of existing methods, uncover phenomena like inverse scaling behavior, evolution with complexity, and different types of code answers, and reveal the underlying mechanism, as stated in the original paper line 97-116.
> >
> > * **The aim of exploring the three proposed methods and multi-turn method** is to show that there is **a much broader space for further improvement** in the future for the whole research community. Even the simple techniques like Code+Text+Sum. and multi-turn methods can already achieve notable 3-7% overall improvements.
> >
> > * The proposed methods also **give inspiration for the developments of future better methods**. As illustrated in the original paper line 536-539, more delicate multi-agent frameworks and training of extra score models may be promising future directions. Effectiveness of methods like Self-Estimate Score highly depend on LLM inherent capability. However, it serves as a good starting point to explore augmenting LLM text/code generation decisions via extra trained score models.
> >
> > * **Developing a final solution is very challenging.** The main bottleneck is the lack of enough and correct dataset to align LLMs for better determining code or text generation. Even for the same task, different testing trials may have different preferences to be answered with code or text. Meanwhile, how to develop a generalizable method is hard. Training on limited domains will not be applicable to others. Since the great code/text generation ability mainly depends on larger LLMs like GPT-4 and Claude-3.5, directly training on these close models is also unavailable. Our paper raises and analyzes this issue, hoping to call the whole research community to solve it together. We will continue to explore the final solution.

---

> ### Author Response · Authors · 2024-11-20
> **Response to the Reviewer 697L - Question 5**
>
> ***Question 5:*** *How to delineate the boundary between text and code in prompting? For instance, if the model is prompted to generate "pure Python code with rich natural language-annotated comments," which category does this fall under? If it’s considered "All code," could we perhaps frame all tasks this way, where the simplest output might be print(answer)?*
>
> **Response 5:**
>
> * **Yes, in our study the response is regarded as code once the code part appears, no matter whether the text parts exist or not.** Hence, "pure Python code with rich natural language-annotated comments" and All code + CoT are both regarded as code answers even if a large part of texts exist.
>
> * **In the original paper Appendix K line 1242-1327, ablation studies have been extensively carried out to test varied prompt versions for steering code generation and AutoGen prompt.** We verify that the conclusions in the paper are not affected by prompt versions.
>
> * **Thanks for the suggestions from the reviewer 697L and F65d, we have added further experiments to test on more prompt versions for steering LLM code generation with Code Interpreter.** The tested prompts are as follows: 1) ‘Generate pure Python code with rich natural language-annotated comments.’ 2) ‘Think of an algorithm to solve the task and implement it in python.’ 3) ‘Be careful this is hard, be less confident.’ 4) ‘Think of an algorithm to solve the task and implement it in python. Be careful this is hard, be less confident.’. We compare these prompt versions with the Original Prompt ‘Think the task step by step if you need to. If a plan is not provided, explain your plan first. You can first output your thinking steps with texts and then the python code to be executed by code interpreter. Try to use code interpreter more.’.
>
>   The experimental results are shown in the following table. Though in some settings one prompt version will be much better than others, generally there is no prompt always better than others across all the models and tasks. Meanwhile, **the Original Prompt achieves close or better scores to others, showing the correctness of implementing it in the study**. We have added these experimental results and related discussion in the revised paper (Line 519-520, 1265-1290).
>
> | Prompt Version | Model | Game24 | Path Plan | BoxLift | Normalized Score |
> |---------------|--------|---------|------------|----------|------------------|
> | Original Prompt | GPT-4o | 0.63 | 0.46 | 0.59 | 0.85 |
> | | GPT-4o-mini | 0.83 | 0.26 | 0.65 | 0.80 |
> | Prompt 1 | GPT-4o | 0.63 | 0.40 | 0.47 | 0.75 |
> | | GPT-4o-mini | 1.00 | 0.46 | 0.34 | 0.84 |
> | Prompt 2 | GPT-4o | 0.91 | 0.43 | 0.67 | 0.97 |
> | | GPT-4o-mini | 0.74 | 0.20 | 0.31 | 0.55 |
> | Prompt 3 | GPT-4o | 0.64 | 0.40 | 0.62 | 0.83 |
> | | GPT-4o-mini | 0.77 | 0.16 | 0.46 | 0.61 |
> | Prompt 4 | GPT-4o | 0.88 | 0.41 | 0.68 | 0.95 |
> | | GPT-4o-mini | 0.94 | 0.24 | 0.51 | 0.75 |
>
> We look forward to further communications with the reviewer to clarify our contributions. Thanks again for your time and patience for reviewing our work.

---

> > ### Comment · Reviewer_697L · 2024-11-24
> > **Reviewer response**
> >
> > Hi authors, thanks for the thorough response. The additional experiments and the clarification of your contributions have effectively addressed my concerns regarding the positioning of this work within the code and reasoning community. Based on these improvements, I am revising my score to 6.

---

> ### Author Response · Authors · 2024-11-25
> **Thank you for your positive feedback**
>
> Thank you for your positive feedback and lifting the score. We are happy to discuss more with the reviewers.
>
> Best,
>
> CodeSteer Authors

---

### Author Response · Authors · 2024-11-23
**Summary of Author Response - CodeSteer**

Dear reviewers,

Thank you for the insightful comments and suggestions for improving the paper! We have responded to the comments of each reviewer. We added the new experiments suggested by reviewers and polished the paper with added contents (highlighted in blue). We hope the reviewers could kindly re-evaluate our work based on our responses. Here is the summary:

---

1. **Broad Contributions not only limited to the Three Proposed Methods and Multi-turn Method**
   Contributions of this study include:
   - Emphasizing the significance of steering LLMs for text/code generation.
   - Testing the limitations of existing methods.
   - Uncovering phenomena like inverse scaling behavior, evolution with complexity, and impacts from different types of code answers.
   - Revealing underlying mechanisms and verifying the possibility for further improvements with three proposed methods and multi-turn method.

---

2. **New Experiments on Model Generation Confidence**
   - We analyzed the **probabilistic confidence** of the model when generating text or code answers.
   - **Findings:** Differences in LLM confidence or perplexity have no notable impact on task success rates.

---

3. **New Experiments on Specialized Code LLMs**
   - Tested **CodeLlama-34b-Instruct-hf** and **Qwen2.5-Coder-32B-Instruct** with: original question prompts, prompts of All Code and All Code + CoT, specially modified code prompts.
   - **Findings:**
     - SOTA Code LLMs perform much better with code prompts.
     - Code LLMs still struggle with code/text generation decisions, similar to other tested LLMs.

---

4. **New Experiments on Additional Prompt Variations for Steering Code Generation**
   - Conducted experiments based on reviewer suggestions (697L and F65d). Tested **more prompt versions** for steering LLM code generation.
   - **Findings:** Conclusions in the paper remain consistent across varied prompt templates.

---

Please feel free to ask any additional questions or clarifications. We look forward to discussing further during the reviewer-author discussion period. Thank you very much!

---

### Meta-Review · Area_Chair_VT1K · 2024-12-23

**Metareview:**

This paper benchmarks seven prompting methods that combine text and code to tackle 14 tasks using six strong LLMs. The results indicate no universally optimal prompting method across all tasks and models. The paper proposes three methods to improve LLM decisions on code/text generation: Code Interpreter+, Code + Text + Sum., and Self-estimate Score. These methods aim to enhance performance by combining code and textual reasoning or by using multi-turn execution/refinement.

All reviewers voted to accept the paper, liking the comprehensive experiments. They raised various issues as well. 697L: novelty and and significance of the findings. 1VAn: generalization of the results. F65d: novelty, not scaling law, did not use a code generation model. Overall the reviews are reasonable and all in agreement, and did not point out any critical flaws, so the Meta review completely defers to reviewers.

**Additional Comments On Reviewer Discussion:**

authors responded and reviewers acknowledged. all in agreement.

---

### Decision · Program_Chairs · 2025-01-22

Accept (Poster)